# 1000 Layer Networks for Self-Supervised RL: Scaling Depth Can Enable New Goal-Reaching Capabilities

**Kevin Wang**
Princeton University
kw6487@princeton.edu

**Ishaan Javali**
Princeton University
ijavali@princeton.edu

**Michał Bortkiewicz**
Warsaw University of Technology
michal.bortkiewicz.dokt@pw.edu.pl

**Tomasz Trzciński**
Warsaw University of Technology,
Tooploox, IDEAS Research Institute

**Benjamin Eysenbach**
Princeton University
eysenbach@princeton.edu

## Abstract

Scaling up self-supervised learning has driven breakthroughs in language and vision, yet comparable progress has remained elusive in reinforcement learning (RL). In this paper, we study building blocks for self-supervised RL that unlock substantial improvements in scalability, with network depth serving as a critical factor. Whereas most RL papers in recent years have relied on shallow architectures (around $2-5$ layers), we demonstrate that increasing the depth up to 1024 layers can significantly boost performance. Our experiments are conducted in an unsupervised goal-conditioned setting, where no demonstrations or rewards are provided, so an agent must explore (from scratch) and learn how to maximize the likelihood of reaching commanded goals. Evaluated on simulated locomotion and manipulation tasks, our approach increases performance on the self-supervised contrastive RL algorithm by $2\times - 50\times$, outperforming other goal-conditioned baselines. Increasing the model depth not only increases success rates but also qualitatively changes the behaviors learned. The project webpage and code can be found here: https://wang-kevin3290.github.io/scaling-crl/.

## 1 Introduction

While scaling model size has been an effective recipe in many areas of machine learning, its role and impact in reinforcement learning (RL) remain unclear. The typical model size for state-based RL tasks is between 2 to 5 layers (Raffin et al., 2021; Huang et al., 2022). In contrast, it is not uncommon to use very deep networks in other domain areas; Llama 3 (Dubey et al., 2024) and Stable Diffusion 3 (Esser et al., 2024) have hundreds of layers. In fields such as vision (Radford et al., 2021; Zhai et al., 2021; Dehghani et al., 2023) and language (Srivastava et al., 2023), models often only acquire the ability to solve certain tasks once they are larger than a critical scale. In the RL setting, many researchers have searched for similar emergent phenomena (Srivastava et al., 2023), but these papers typically report only small marginal benefits and typically only on tasks where small models already achieve some degree of success (Nauman et al., 2024b; Lee et al., 2024; Farebrother et al., 2024). A key open question in RL today is whether it is possible to achieve similar jumps in performance by scaling RL networks.

39th Conference on Neural Information Processing Systems (NeurIPS 2025).

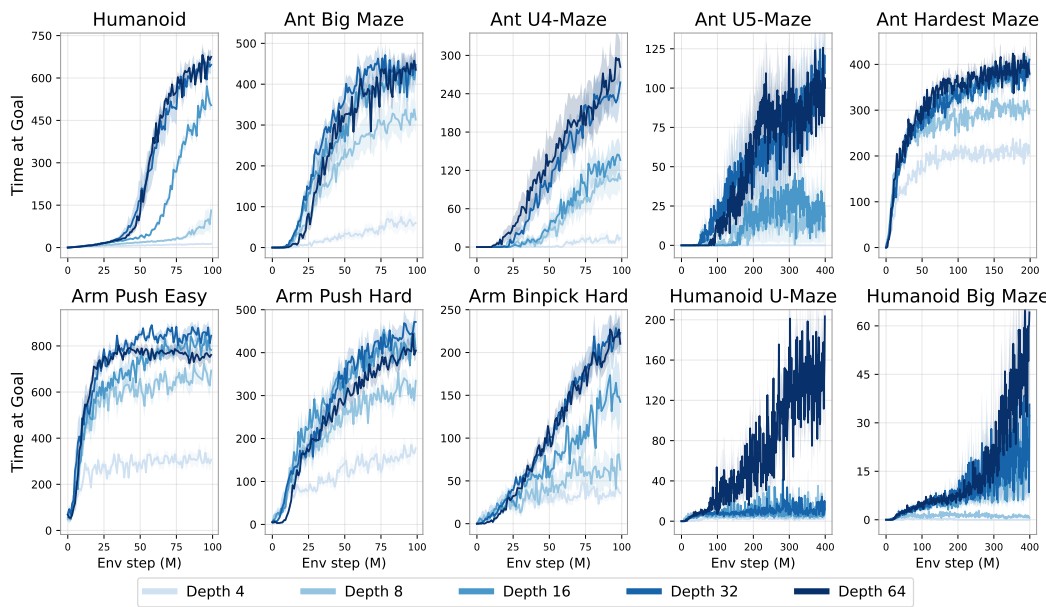

Figure 1: **Scaling network depth yields performance gains** across a suite of locomotion, navigation, and manipulation tasks, ranging from doubling performance to 50× improvements on Humanoid-based tasks. Notably, rather than scaling smoothly, performance often jumps at specific critical depths (e.g., 8 layers on Ant Big Maze, 64 on Humanoid U-Maze), which correspond to the emergence of qualitatively distinct policies (see Section 4).

At first glance, it makes sense why training very large RL networks should be difficult: the RL problem provides very few bits of feedback (e.g., only a sparse reward after a long sequence of observations), so the ratio of feedback to parameters is very small. The conventional wisdom (LeCun, 2016), reflected in many recent models (Radford, 2018; Chen et al., 2020; Goyal et al., 2019), has been that large AI systems must be trained primarily in a self-supervised fashion and that RL should only be used to finetune these models. Indeed, many of the recent breakthroughs in other fields have been primarily achieved with *self-supervised* methods, whether in computer vision (Caron et al., 2021; Radford et al., 2021; Liu et al., 2024), NLP (Srivastava et al., 2023), or multimodal learning (Zong et al., 2024). Thus, if we hope to scale reinforcement learning methods, self-supervision will likely be a key ingredient.

In this paper, we will study *building blocks* for scaling reinforcement learning. Our first step is to rethink the conventional wisdom above: "reinforcement learning" and "self-supervised learning" are not diametric learning rules, but rather can be married together into self-supervised RL systems that explore and learn policies without reference to a reward function or demonstrations (Eysenbach et al., 2021, 2022; Lee et al., 2022). In this work, we use one of the simplest self-supervised RL algorithms, contrastive RL (CRL) (Eysenbach et al., 2022). The second step is to recognize the importance of increasing data availability. We will do this by building on recent GPU-accelerated RL frameworks (Makoviychuk et al., 2021; Rutherford et al., 2023; Rudin et al., 2022; Bortkiewicz et al., 2024). The third step is to increase network depth, using networks that are up to 100× deeper than those typically found in prior work. Stabilizing the training of such networks will require incorporating architectural techniques from prior work, including residual connections (He et al., 2015), layer normalization (Ba et al., 2016), and Swish activation (Ramachandran et al., 2018). Our experiments will also study the relative importance of batch size and network width.

The primary contribution of this work is to show that a method that integrates these building blocks into a single RL approach exhibits strong scalability:

- **Empirical Scalability:** We observe a significant performance increase, more than 20× in half of the environments and outperforming other standard goal-conditioned baselines. These performance gains correspond to qualitatively distinct policies that emerge as the scale increases.
- **Scaling Depth in Network Architecture:** While many prior RL works have primarily focused on increasing network width, they often report limited or even negative returns

when expanding depth (Lee et al., 2024; Nauman et al., 2024b). In contrast, our approach unlocks the ability to scale along the axis of depth, yielding performance improvements that surpass those from scaling width alone (see Sec. 4).

- **Empirical Analysis**: We conduct an extensive analysis of the key components in our scaling approach, uncovering critical factors and offering new insights.

We anticipate that future research may build on this foundation by uncovering additional building blocks.

## 2   Related Work

Natural Language Processing (NLP) and Computer Vision (CV) have recently converged in adopting similar architectures (i.e. transformers) and shared learning paradigms (i.e self-supervised learning), which together have enabled transformative capabilities of large-scale models (Vaswani et al., 2017; Srivastava et al., 2023; Zhai et al., 2021; Dehghani et al., 2023; Wei et al., 2022). In contrast, achieving similar advancements in reinforcement learning (RL) remains challenging. Several studies have explored the obstacles to scaling large RL models, including parameter underutilization (Obando-Ceron et al., 2024), plasticity and capacity loss (Lyle et al., 2024, 2022), data sparsity (Andrychowicz et al., 2017; LeCun, 2016), and training instabilities (Ota et al., 2021; Henderson et al., 2018; Van Hasselt et al., 2018; Nauman et al., 2024a). As a result, current efforts to scale RL models are largely restricted to specific problem domains, such as imitation learning (Tuyls et al., 2024), multi-agent games (Neumann and Gros, 2022), language-guided RL (Driess et al., 2023; Ahn et al., 2022), and discrete action spaces (Obando-Ceron et al., 2024; Schwarzer et al., 2023).

Recent approaches suggest several promising directions, including new architectural paradigms (Obando-Ceron et al., 2024), distributed training approaches (Ota et al., 2021; Espeholt et al., 2018), distributional RL (Kumar et al., 2023), and distillation (Team et al., 2023). Compared to these approaches, our method makes a simple extension to an existing self-supervised RL algorithm. The most recent works in this vein include Lee et al. (2024) and Nauman et al. (2024b), which leverage residual connections to facilitate the training of wider networks. These efforts primarily focus on network width, noting limited gains from additional depth, thus both works use architectures with only four MLP layers. In our method, we find that scaling width indeed improves performance (Section 4.4); however, our approach also enables scaling along depth, proving to be more powerful than width alone.

One notable effort to train deeper networks is described by Farebrother et al. (2024), who cast value-based RL into a classification problem by discretizing the TD objective into a categorical cross-entropy loss. This approach draws on the conjecture that classification-based methods can be more robust and stable and thus may exhibit better scaling properties than their regressive counterparts (Torgo and Gama, 1996; Farebrother et al., 2024). The CRL algorithm that we use effectively uses a cross-entropy loss as well (Eysenbach et al., 2022). Its InfoNCE objective is a generalization of the cross-entropy loss, thereby performing RL tasks by effectively classifying whether current states and actions belong to the same or different trajectory that leads toward a goal state. In this vein, our work serves as a second piece of evidence that classification, much like cross-entropy's role in the scaling success in NLP, could be a potential building block in RL.

## 3   Preliminaries

This section introduces notation and definitions for goal-conditioned RL and contrastive RL. Our focus is on online RL, where a replay buffer stores the most recent trajectories, and the critic is trained in a self-supervised manner.

**Goal-Conditioned Reinforcement Learning**   We define a goal-conditioned MDP as tuple $\mathcal{M}_g = (\mathcal{S}, \mathcal{A}, p_0, p, p_g, r_g, \gamma)$, where the agent interacts with the environment to reach arbitrary goals (Kaelbling, 1993; Andrychowicz et al., 2017; Blier et al., 2021). At every time step $t$, the agent observes state $s_t \in \mathcal{S}$ and performs a corresponding action $a_t \in \mathcal{A}$. The agent starts interaction in states sampled from $p_0(s_0)$, and the interaction dynamics are defined by the transition probability distribution $p(s_{t+1} \mid s_t, a_t)$. Goals $g \in \mathcal{G}$ are defined in a goal space $\mathcal{G}$, which is related to $\mathcal{S}$ via a mapping $f : \mathcal{S} \to \mathcal{G}$. For example, $\mathcal{G}$ may correspond to a subset of state dimensions. The prior distribution

over goals is defined by $p_g(g)$. The reward function is defined as the probability density of reaching the goal in the next time step $r_g(s_t, a_t) \triangleq (1 - \gamma)p(s_{t+1} = g \mid s_t, a_t)$, with discount factor $\gamma$.

In this setting, the goal-conditioned policy $\pi(a \mid s, g)$ receives both the current observation of the environment as well as a goal. We define the discounted state visitation distribution as $p_\gamma^{\pi(\cdot|\cdot,g)}(s) \triangleq (1 - \gamma) \sum_{t=0}^{\infty} \gamma^t p_t^{\pi(\cdot|\cdot,g)}(s)$, where $p_t^\pi(s)$ is the probability that policy $\pi$ visits $s$ after exactly $t$ steps, when conditioned with $g$. This last expression is precisely the $Q$-function of the policy $\pi(\cdot \mid \cdot, g)$ for the reward $r_g$: $Q_g^\pi(s, a) \triangleq p_\gamma^{\pi(\cdot|\cdot,g)}(g \mid s, a)$. The objective is to maximize the expected reward:

$$\max_\pi \mathbb{E}_{p_0(s_0), p_g(g), \pi(\cdot|\cdot,g)} \left[ \sum_{t=0}^{\infty} \gamma^t r_g(s_t, a_t) \right]. \tag{1}$$

**Contrastive Reinforcement Learning.** Our experiments will use the contrastive RL algorithm (Eysenbach et al., 2022) to solve goal-conditioned problems. Contrastive RL is an actor-critic method; we will use $f_{\phi,\psi}(s, a, g)$ to denote the critic and $\pi_\theta(a \mid s, g)$ to denote the policy. The critic is parametrized with two neural networks that return state, action pair embedding $\phi(s, a)$ and goal embedding $\psi(g)$. The critic's output is defined as the $l^2$-norm between these embeddings: $f_{\phi,\psi}(s, a, g) = \|\phi(s, a) - \psi(g)\|_2$. The critic is trained with the InfoNCE objective (Sohn, 2016) as in previous works (Eysenbach et al., 2022, 2021; Zheng et al., 2023, 2024; Myers et al., 2024; Bortkiewicz et al., 2024). Training is conducted on batches $\mathcal{B}$, where $s_i, a_i, g_i$ represent the state, action, and goal (future state) sampled from the same trajectory, while $g_j$ represents a goal sampled from a different, random trajectory. The objective function is defined as:

$$\min_{\phi,\psi} \mathbb{E}_{\mathcal{B}} \left[ -\sum_{i=1}^{|\mathcal{B}|} \log \left( \frac{e^{f_{\phi,\psi}(s_i, a_i, g_i)}}{\sum_{j=1}^{K} e^{f_{\phi,\psi}(s_i, a_i, g_j)}} \right) \right].$$

The policy $\pi_\theta(a \mid s, g)$ is trained to maximize the critic:

$$\max_{\pi_\theta} \mathbb{E}_{\substack{p_0(s_0), p(s_{t+1}|s_t, a_t), \\ p_g(g), \pi_\theta(a|s,g)}} [f_{\phi,\psi}(s, a, g)].$$

**Residual Connections** We incorporate residual connections (He et al., 2015) into our architecture, following their successful use in RL (Farebrother et al., 2024; Lee et al., 2024; Nauman et al., 2024b). A residual block transforms a given representation $\mathbf{h}_i$ by adding a learned residual function $F_i(\mathbf{h}_i)$ to the original representation. Mathematically, this is expressed as:

$$\mathbf{h}_{i+1} = \mathbf{h}_i + F_i(\mathbf{h}_i)$$

where $\mathbf{h}_{i+1}$ is the output representation, $\mathbf{h}_i$ is the input representation, and $F_i(\mathbf{h}_i)$ is a transformation learned through the network (e.g., using one or more layers). The addition ensures that the network learns modifications to the input rather than entirely new transformations, helping to preserve useful features from earlier layers. Residual con-

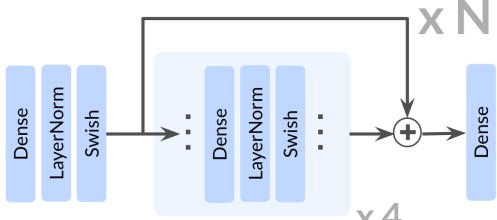

Figure 2: **Architecture.** Our approach integrates residual connections into both the actor and critic networks of the Contrastive RL algorithm. The depth of this residual architecture is defined as the total number of Dense layers across the residual blocks, which, with our residual block size of 4, equates to $4N$.

nections improve gradient propagation by introducing shortcut paths (He et al., 2016; Veit et al., 2016), enabling more effective training of deep models.

## 4 Experiments

### 4.1 Experimental Setup

**Environments.** All RL experiments use the JaxGCRL codebase (Bortkiewicz et al., 2024), which facilitates fast online GCRL experiments based on Brax (Freeman et al., 2021) and MJX (Todorov

et al., 2012) environments. The specific environments used are a range of locomotion, navigation, and robotic manipulation tasks, for details see Appendix B. We use a sparse reward setting, with $r = 1$ only when the agent is in the goal proximity. For evaluation, we measure the number of time steps (out of 1000) that the agent is near the goal. When reporting an algorithm's performance as a single number, we compute the average score over the last five epochs of training.

**Architectural Components**   We employ residual connections from the ResNet architecture (He et al., 2015), with each residual block consisting of four repeated units of a Dense layer, a Layer Normalization (Ba et al., 2016) layer, and Swish activation (Ramachandran et al., 2018). We apply the residual connections immediately following the final activation of the residual block, as shown in Figure 2. In this paper, we define the depth of the network as the total number of Dense layers across all residual blocks in the architecture. In all experiments, the depth refers to the configuration of the actor network and both critic encoder networks, which are scaled jointly, except for the ablation experiment in Section 4.4.

## 4.2   Scaling Depth in Contrastive RL

We start by studying how increasing network depth can increase performance. Both the JaxGCRL benchmark and relevant prior work (Lee et al., 2024; Nauman et al., 2024b; Zheng et al., 2024) use MLPs with a depth of 4, and as such we adopt it as our baseline. In contrast, we will study networks of depth 8, 16, 32, and 64. The results in Figure 1 demonstrate that deeper networks achieve significant performance improvements across a diverse range of locomotion, navigation, and manipulation tasks. Compared to the 4-layer models typical in prior work, deeper networks achieve $2 - 5\times$ gains in robotic manipulation tasks, over $20\times$ gains in long-horizon maze tasks such as Ant U4-Maze and Ant U5-Maze, and over $50\times$ gains in humanoid-based tasks. The full table of performance increases up to depth 64 is provided in Table 1.

In Figure 12, we present results the same 10 environments, but compared against SAC, SAC+HER, TD3+HER, GCBC, and GCSL. Scaling CRL leads to substantial performance improvements, outperforming all other baselines in 8 out of 10 tasks. The only exception is SAC on the Humanoid Maze environments, where it exhibits greater sample efficiency early on; however, scaled CRL eventually reaches comparable performance. These results highlight that scaling the depth of the CRL algorithm enables state-of-the-art performance in goal-conditioned reinforcement learning.

## 4.3   Emergent Policies Through Depth

A closer examination of the results from the performance curves in Figure 1 reveals a notable pattern: instead of a gradual improvement in performance as depth increases, there are pronounced jumps that occur once a *critical depth* threshold is reached (also shown in Figure 5). The critical depths vary by environment, ranging from 8 layers (e.g. Ant Big Maze) to 64 layers in the Humanoid U-Maze task, with further jumps occurring even at depths of 1024 layers (see the Testing Limits section, Section 4.4).

Prompted by this observation, we visualized the learned policies at various depths and found qualitatively distinct skills and behaviors exhibited. This is particularly pronounced in the humanoid-based tasks, as illustrated in Figure 3. Networks with a depth of 4 exhibit rudimentary policies where the agent either falls or throws itself toward the target. Only at a critical depth

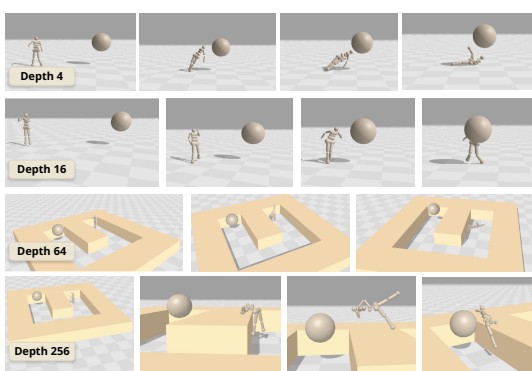

Figure 3: **Increasing depth results in new capabilities: Row 1**: A depth-4 agent collapses and throws itself toward the goal. **Row 2**: A depth-16 agent walks upright. **Row 3**: A depth-64 agent struggles and falls. **Row 4**: A depth-256 agent vaults the wall acrobatically.

of 16 does the agent develop the ability to walk upright into the goal. In the Humanoid U-Maze environment, networks of depth 64 struggle to navigate around the intermediary wall, collapsing on the ground. Remarkably at a depth of 256, the agent learns unique behaviors on Humanoid U-Maze. These behaviors include folding forward into a leveraged position to propel itself over walls and

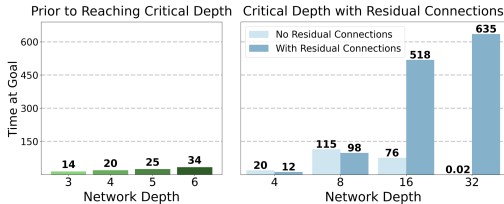
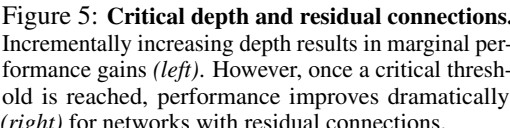
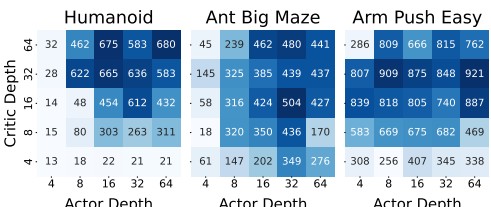

Figure 5: **Critical depth and residual connections.** Incrementally increasing depth results in marginal performance gains *(left)*. However, once a critical threshold is reached, performance improves dramatically *(right)* for networks with residual connections.

Figure 6: **Actor vs. Critic.** In Arm Push Easy, scaling the critic is more effective; in Ant Big Maze, the actor matters more. For Humanoid, scaling both is necessary. These results suggest that actor and critic scaling can complement each other for CRL.

shifting into a seated posture over the intermediary obstacle to worm its way toward the goal (one of these policies is illustrated in the fourth row of Figure 3). To the best of our knowledge, this is the first goal-conditioned approach to document such behaviors on the humanoid environment.

### 4.4 What Matters for CRL Scaling

**Width vs. Depth** Past literature has shown that scaling network width can be effective (Lee et al., 2024; Nauman et al., 2024b). In Figure 4, we find that scaling width is also helpful in our experiments: wider networks consistently outperform narrower networks (depth held constant at 4). However, depth seems to be a more effective axis for scaling: simply doubling the depth to 8 (width held constant at 256) outperforms the widest networks in all three environments. The advantage of depth scaling is most pronounced in the Humanoid environment (observation dimension 268), followed by Ant Big Maze (dimension 29) and Arm Push Easy (dimension 17), suggesting that the comparative benefit may increase with higher observation dimensionality.

Note additionally that the parameter count scales linearly with width but quadratically with depth. For comparison, a network with 4 MLP layers

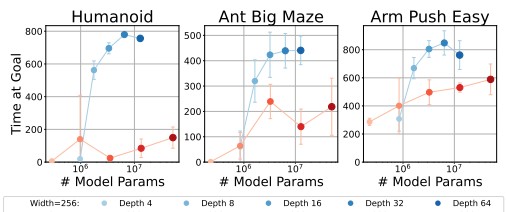

Figure 4: **Scaling network width vs. depth**. Here, we reflect findings from previous works (Lee et al., 2024; Nauman et al., 2024b) which suggest that increasing network width can enhance performance. However, in contrast to prior work, our method is able to scale depth, yielding more impactful performance gains. For instance, in the Humanoid environment, raising the width to 2048 (depth=4) fails to match the performance achieved by simply doubling the depth to 8 (width=256). The comparative advantage of scaling depth is more pronounced as the observational dimensionality increases.

and 2048 hidden units has roughly 35M parameters, while one with a depth of 32 and 256 hidden units has only around 2M. Therefore, when operating under a fixed FLOP compute budget or specific memory constraints, depth scaling may be a more computationally efficient approach to improving network performance.

**Scaling the Actor vs. Critic Networks** To investigate the role of scaling in the actor and critic networks, Figure 6 presents the final performance for various combinations of actor and critic depths across three environments. Prior work (Nauman et al., 2024b; Lee et al., 2024) focuses on scaling the critic network, finding that scaling the actor degrades performance. In contrast, while we do find that scaling the critic is more impactful in two of the three environments (Humanoid, Arm Push Easy), our method benefits from scaling the actor network jointly, with one environment (Ant Big Maze) demonstrating actor scaling to be more impactful. Thus, our method suggests that scaling both the actor and critic networks can play a complementary role in enhancing performance.

**Deep Networks Unlock Batch Size Scaling** Scaling batch size has been well-established in other areas of machine learning (Chen et al., 2022; Zhang et al., 2024). However, this approach has not translated as effectively to reinforcement learning (RL), and prior work has even reported negative impacts on value-based RL (Obando-Ceron et al., 2023). Indeed, in our experiments,

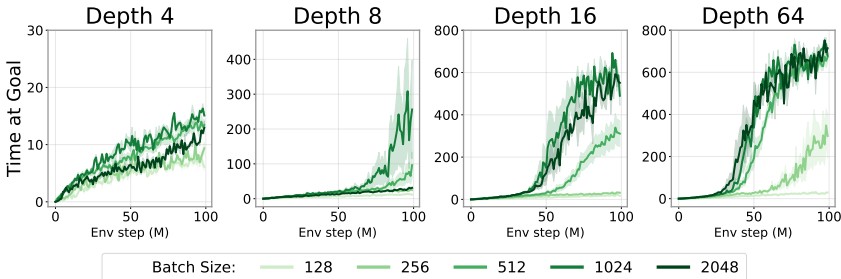

Figure 7: **Deeper networks unlock batch size scaling.** We find that as depth increases from 4 to 64 in Humanoid, larger networks can effectively leverage batch size scaling to achieve further improvements.

simply increasing the batch size for the original CRL networks yields only marginal differences in performance (Figure 7, top left).

At first glance, this might seem counterintuitive: since reinforcement learning typically involves fewer informational bits per piece of training data (LeCun, 2016), one might expect higher variance in batch loss or gradients, suggesting the need for larger batch sizes to compensate. At the same time, this possibility hinges on whether the model in question can actually make use of a bigger batch size—in domains of ML where scaling has been successful, larger batch sizes usually bring the most benefit when coupled with sufficiently large models (Zhang et al., 2024; Chen et al., 2022). One hypothesis is that the small models traditionally used in RL may obscure the underlying benefits of larger batch size.

To test this hypothesis, we study the effect of increasing the batch size for networks of varying depths. As shown in Figure 7, scaling the batch size becomes effective as network depth grows. This finding offers evidence that by scaling network capacity, we may simultaneously unlock the benefits of larger batch size, potentially making it an important component in the broader pursuit of scaling self-supervised RL.

**Training Contrastive RL with 1000+ Layers**  We next study whether further increasing depth beyond 64 layers further improves performance. We use the Humanoid maze tasks as these are both the most challenging environments in the benchmark and also seem to benefit from the deepest scaling. The results, shown in Figure 12, indicate that performance continues to substantially improve as network depth reaches 256 and 1024 layers in the Humanoid U-Maze environment. While we were unable to scale beyond 1024 layers due to computational constraints, we expect to see continued improvements with even greater depths, especially on the most challenging tasks.

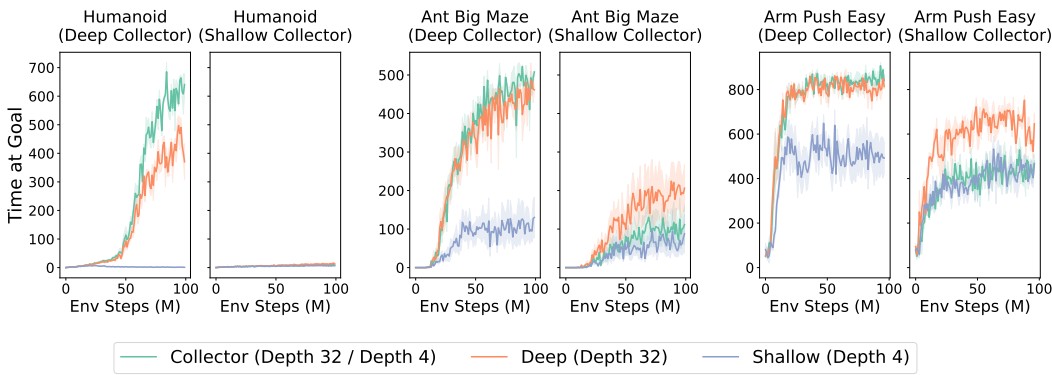

Figure 8: We disentangle the effects of exploration and expressivity on depth scaling by training three networks in parallel: a "collector," plus one deep and one shallow learner that train only from the collector's shared replay buffer. In all three environments, when using a deep collector (i.e. good data coverage), the deep learner outperforms the shallow learner, indicating that expressivity is crucial when controlling for good exploration. With a shallow collector (poor exploration), even the deep learner cannot overcome the limitations of insufficient data coverage. As such, the benefits of depth scaling arise from a combination of improved exploration and increased expressivity working jointly.

## 4.5 Why Scaling Happens

**Depth Enhances Contrastive Representations**
The long-horizon setting has been a long-standing
challenge in RL particularly in unsupervised goal-
conditioned settings where there is no auxiliary re-
ward feedback (Gupta et al., 2019). The family of
U-Maze environments requires a global understand-
ing of the maze layout for effective navigation. We
consider a variant of the Ant U-Maze environment,
the U4-maze, in which the agent must initially move
in the direction opposite the goal to loop around and
ultimately reach it. As shown in Figure 9, we ob-
serve a qualitative difference in the behavior of the
shallow network (depth 4) compared to the deep net-
work (depth 64). The visualized Q-values computed
from the critic encoder representations reveal that
the depth 4 network seemingly relies on Euclidean
distance to the goal as a proxy for the Q value, even
when a wall obstructs the direct path. In contrast, the
depth 64 critic network learns richer representations,
enabling it to effectively capture the topology of the
maze as visualized by the trail of high Q values along
the inner edge. These findings suggest that increasing
network depth leads to richer learned representations,
enabling deeper networks to better capture environment topology and achieve more comprehensive
state-space coverage in a self-supervised manner.

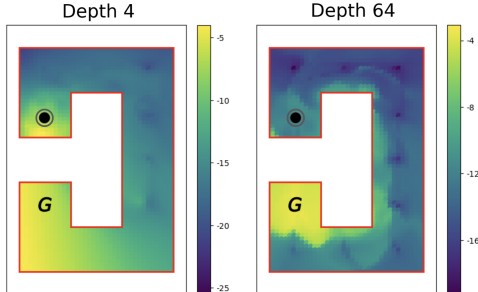

Figure 9: **Deeper Q-functions are qualitatively
different.** In the U4-Maze, the start and goal posi-
tions are indicated by the ⊙ and **G** symbols respec-
tively, and the visualized Q values are computed
via the $L_2$ distance in the learned representation
space, i.e., $Q(s, a, g) = \|\phi(s, a) - \psi(g)\|_2$. The
shallow depth 4 network *(left)* naively relies on
Euclidean proximity, showing high Q values near
the start despite a maze wall. In contrast, the depth
64 network *(right)* clusters high Q values at the
goal, gradually tapering along the interior.

**Depth Enhances Exploration and Expressivity in a Synergized Way**  Our earlier results suggested
that deeper networks achieve greater state-action coverage. To better understand why scaling works,
we sought to determine to whether improved data alone explains the benefits of scaling, or whether it
acts in conjunction with other factors. Thus, we designed an experiment in Figure 8 in which we train
three networks in parallel: one network, the "collector," interacts with the environment and writes
all experience to a shared replay buffer. Alongside it, two additional "learners", one deep and one
shallow, train concurrently. Crucially, these two learners never collect their own data; they train only
from the collector's buffer. This design holds the data distribution constant while varying the model's
capacity, so any performance gap between the deep and shallow learners must come from expressivity
rather than exploration. When the collector is deep (e.g., depth 32), across all three environments
the deep learner substantially outperforms the shallow one across all three environments, indicating
that the expressivity of the deep networks is critical. On the other hand, we repeat the experiment
with shallow collectors (e.g., depth 4), which explores less effectively and therefore populates the
buffer with low-coverage experience. Here, both the deep and shallow learners struggle and achieve
similarly poor performance, which indicates that the deep network's additional capacity does not
overcome the limitations of insufficient data coverage. As such, scaling depth enhances exploration
and expressivity in a synergized way: stronger learning capacity drives more extensive exploration,
and strong data coverage is essential to fully realize the power of stronger learning capacity. Both
aspects jointly contribute to improved performance.

**Deep Networks Learn to Allocate Greater Representational Capacity to States Near the Goal**
In Figure 10 we take a successful trajectory in the Humanoid environment and visualize the embed-
dings of state-action encoder along this trajectory for both deep vs. shallow networks. While the
shallow network (Depth 4) tends to cluster near-goal states tightly together, the deep network produces
more "spread out" representations. This distinction is important: in a self-supervised setting, we want
our representations to separate states that matter—particularly future or goal-relevant states—from
random ones. As such, we want to allocate more representational capacity to such critical regions.
This suggests that deep networks may learn to allocate representational capacity more effectively to
state regions that matter most for the downstream task.

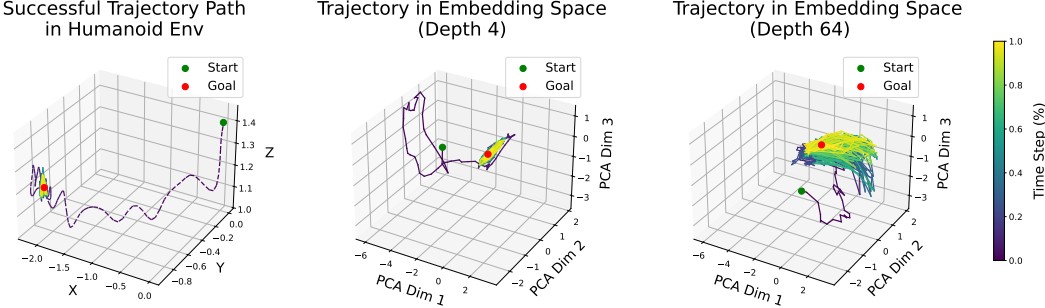

Figure 10: We visualize state-action embeddings from shallow (depth 4) and deep (depth 64) networks along a successful trajectory in the Humanoid task. Near the goal, embeddings from the deep network expand across a curved surface, while those from the shallow network form a tight cluster. This suggests that deeper networks may devote greater representational capacity to regions of the state space that are more frequently visited and play a more critical role in successful task completion.

**Deeper Networks Enable Partial Experience Stitching** Another key challenge in reinforcement learning is learning policies that can generalize to tasks unseen during training. To evaluate this setting, we designed a modified version of the Ant U-Maze environment. As shown in Figure 11 (top right), the original JaxGCRL benchmark assesses the agent's performance on the three farthest goal positions located on the opposite side of the wall. However, instead of training on all possible subgoals (a superset of the evaluation state-goal pairs), we modified the setup to train on start-goal pairs that are at most 3 units apart, ensuring that none of the evaluation pairs ever appear in the training set. Figure 11 demonstrates that depth 4 networks show limited generalization, solving only the easiest goal (4 units away from the start). Depth 16 networks achieve moderate success, while depth 64 networks excel, sometimes solving the most challenging goal position. These results suggest that the increasing network depth results in some degree of stitching, combining ≤3-unit pairs to navigate the 6-unit span of the U-Maze.

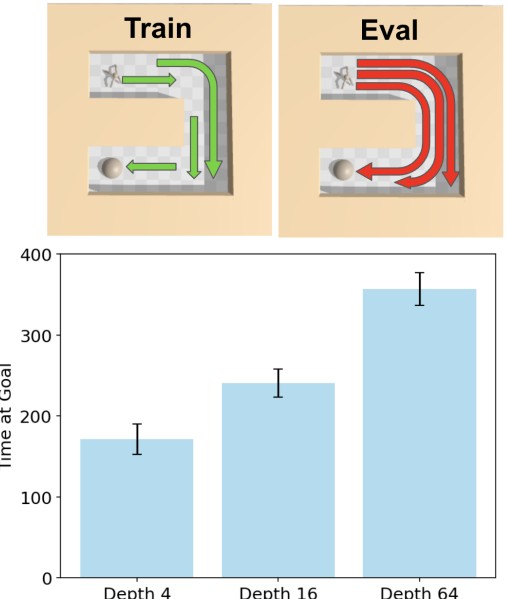

Figure 11: **Deeper networks exhibit improved generalization.** *(Top left)* We modify the training setup of the Ant U-Maze environment such that start-goal pairs are separated by ≤ 3 units. This design guarantees that no evaluation pairs *(Top right)* were encountered during training, testing the ability for combinatorial generalization via stitching. *(Bottom)* Generalization ability improves as network depth grows from 4 to 16 to 64 layers.

**The (CRL) Algorithm is Key** In Appendix A, we show that scaled CRL outperforms other baseline goal-conditioned algorithms and advance the SOTA for goal-conditioned RL. We observe that for temporal difference methods (SAC, SAC+HER, TD3+HER), the performance saturates for networks of depth 4, and there is either zero or negative performance gains from deeper networks. This is in line with previous research showing that these methods benefit mainly from width (Lee et al., 2024; Nauman et al., 2024b). These results suggest that the self-supervised CRL algorithm is critical.

We also experiment with scaling more self-supervised algorithms, namely Goal-Conditioned Behavioral Cloning (GCBC) and Goal-Conditioned Supervised Learning (GCSL). While these methods yield zero success in certain environments, they show some utility in arm manipulation tasks. Interestingly, even a very simple self-supervised algorithm like GCBC benefits from increased depth. This

points to a promising direction for future work of further investigating other self-supervised methods to uncover potentially different or complementary recipes for scaling self-supervised RL.

Finally, recent work has augmented goal-conditioned RL with quasimetric architectures, leveraging the fact that temporal distances satisfy a triangle inequality–based invariance. In Appendix A, we also investigate whether the depth scaling effect persists when applied to these quasimetric networks.

### 4.6 Does Depth Scaling Improve Offline Contrastive RL?

In preliminary experiments, we evaluated depth scaling in the offline goal-conditioned setting using OGBench (Park et al., 2024). We found little evidence that increasing the network depth of CRL improves performance in this offline setting. To further investigate this, we conducted ablations: *(1)* scaling critic depth while holding the actor at 4 or 8 layers, and *(2)* applying cold initialization to the final layers of the critic encoders (Zheng et al., 2024). In all cases, baseline depth 4 networks often had the highest success. A key direction for future work is to see if our method can be adapted to enable scaling in the offline setting.

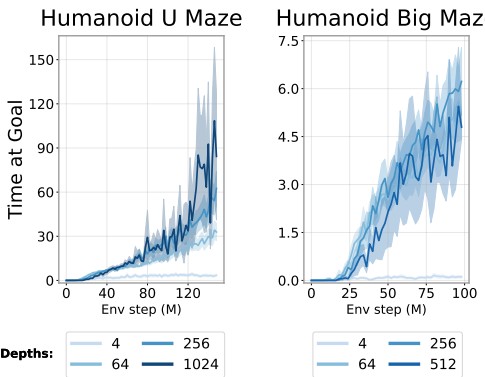

Figure 12: **Testing the limits of scale.** We extend the results from Figure 1 by scaling networks even further on the challenging Humanoid maze environments. We observe continued performance improvements with network depths of 256 and 1024 layers on Humanoid U-Maze. Note that for the 1024-layer networks, we observed the actor loss exploding at the onset of training, so we maintained the actor depth at 512 while using 1024-layer networks only for the two critic encoders.

## 5 Conclusion

Arguably, much of the success of vision and language models today is due to the emergent capabilities they exhibit from scale (Srivastava et al., 2023), leading to many systems reducing the RL problem to a vision or language problem. A critical question for large AI models is: where does the data come from? Unlike supervised learning paradigms, RL methods inherently address this by jointly optimizing both the model and the data collection process through exploration. Ultimately, determining effective ways of building RL systems that demonstrate emergent capabilities may be important for transforming the field into one that trains its own large models. We believe that our work is a step towards these systems. By integrating key components for scaling up RL into a single approach, we show that model performance consistently improves as scale increases in complex tasks. In addition, deep models exhibit qualitatively better behaviors which might be interpreted as implicitly acquired skills necessary to reach the goal.

**Limitations.** The primary limitations of our results are that scaling network depth comes at the cost of compute. An important direction for future work is to study how distributed training might be used to leverage even more compute, and how techniques such as pruning and distillation might be used to decrease the computational costs.

**Impact Statement** This paper presents work whose goal is to advance the field of Machine Learning. There are many potential societal consequences of our work, none which we feel must be specifically highlighted here.

**Acknowledgments.** We gratefully acknowledge Nathaniel Chen, Galen Collier, and the full staff of Princeton Research Computing for their invaluable assistance. We also thank Colin Lu for his discussions and contributions to this work. This research was also partially supported by the National Science Centre, Poland (grant no. 2023/51/D/ST6/01609); the Princeton Laboratory for Artificial Intelligence under Award 2025-97; and the Warsaw University of Technology through the Excellence Initiative: Research University (IDUB) program. Finally, we would also like to thank Jens Tuyls and Harshit Sikchi for providing helpful commends and feedback on the manuscript.

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

# A Additional Experiments

## A.1 Scaled CRL Outperforms All Other Baselines on 8 out of 10 Environments

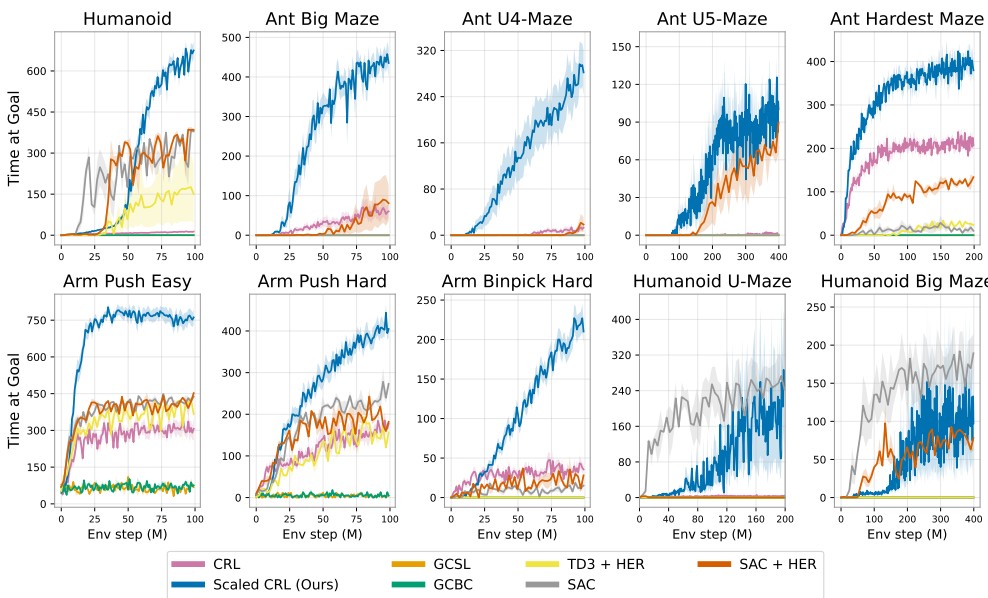

Figure 12: Scaled CRL (Ours) outperforms baselines CRL (original), SAC, SAC+HER, TD3+HER, GCSL, and GCBC in 8 out 10 environments.

In Figure 1, we demonstrated that increasing the depth of the CRL algorithm leads to significant performance improvements over the original CRL (see also Table 1). Here, we show that these gains translate to state-of-the-art results in online goal-conditioned RL, with Scaled CRL outperforming both standard TD-based methods such as SAC, SAC+HER, and TD3+HER, as well as self-supervised imitation-based approaches like GCBC and GCSL.

## A.2 The CRL Algorithm is Key: Depth Scaling is Not Effective on Other Baselines

Next, we investigate whether increasing network depth in the baseline algorithms yields similar performance improvements as observed in CRL. We find that SAC, SAC+HER, and TD3+HER do not benefit from depths beyond four layers, which is consistent with prior findings (Lee et al., 2024; Nauman et al., 2024b). Additionally, GCSL and GCBC fail to achieve any meaningful performance on the Humanoid and Ant Big Maze tasks. Interestingly, we do observe one exception, as GCBC exhibits improved performance with increased depth in the Arm Push Easy environment.

Table 1: Increasing network depth (depth $D = 4 \to 64$) increases performance on CRL (Figure 1). Scaling depth exhibits the greatest benefits on tasks with the largest observation dimension (*Dim*).

| Task | Dim | $D = 4$ | $D = 64$ | Imprv. |
|---|---|---|---|---|
| Arm Binpick Hard | | $38 _{\pm 4}$ | $219 _{\pm 15}$ | $5.7\times$ |
| Arm Push Easy | *17* | $308 _{\pm 33}$ | $762 _{\pm 30}$ | $2.5\times$ |
| Arm Push Hard | | $171 _{\pm 11}$ | $410 _{\pm 13}$ | $2.4\times$ |
| Ant U4-Maze | | $11.4 _{\pm 4.1}$ | $286 _{\pm 36}$ | $25\times$ |
| Ant U5-Maze | *29* | $0.97 _{\pm 0.7}$ | $61 _{\pm 18}$ | $63\times$ |
| Ant Big Maze | | $61 _{\pm 20}$ | $441 _{\pm 25}$ | $7.3\times$ |
| Ant Hardest Maze | | $215 _{\pm 8}$ | $387 _{\pm 21}$ | $1.8\times$ |
| Humanoid | | $12.6 _{\pm 1.3}$ | $649 _{\pm 19}$ | $52\times$ |
| Humanoid U-Maze | *268* | $3.2 _{\pm 1.2}$ | $159 _{\pm 33}$ | $50\times$ |
| Humanoid Big Maze | | $0.06 _{\pm 0.04}$ | $59 _{\pm 21}$ | $1051\times$ |

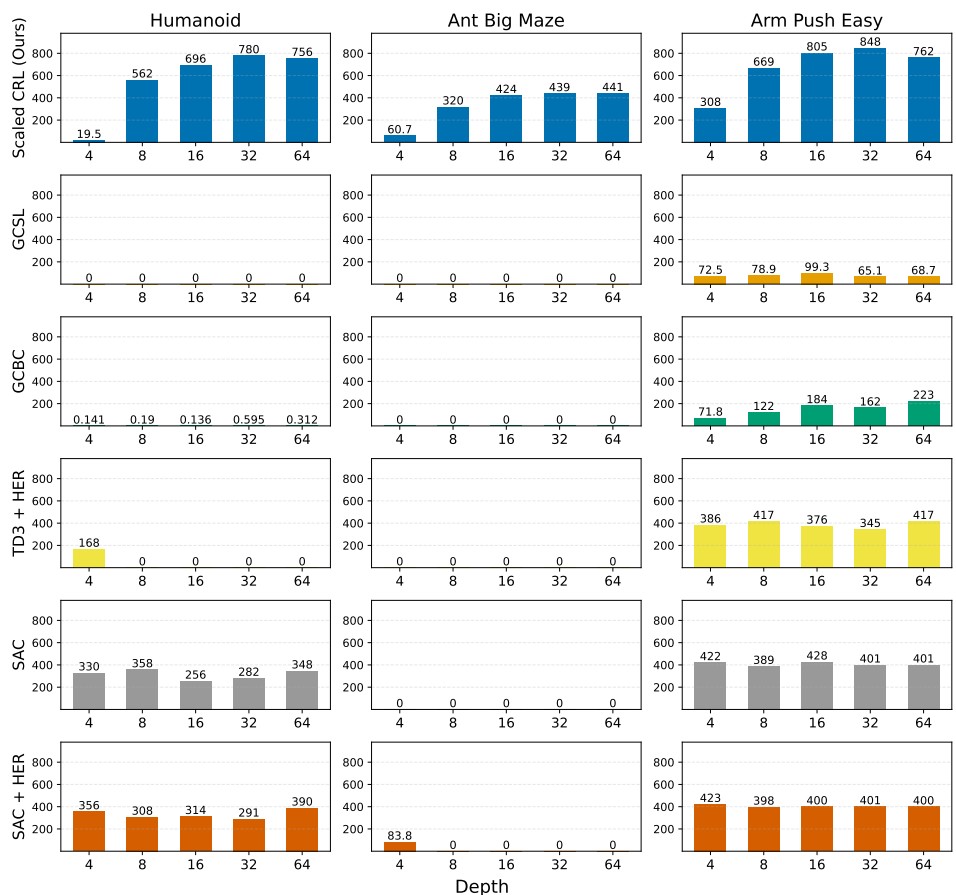

Figure 13: Depth scaling yields limited gains for SAC, SAC+HER, TD3+HER, GCSL, and GCBC.

## A.3 Additional Scaling Experiments: Offline GCBC, BC, and QRL

We further investigate several additional scaling experiments. As shown in Figure 14, our approach successfully scales with depth in the offline GCBC setting on the *antmaze-medium-stitch* task from OGBench. We find that our the combination of layer normalization, residual connections, and Swish activations is critical, suggesting that our architectural choices may be applied to unlock depth scaling in other algorithms and settings. We also attempt to scale depth for behavioral cloning and the QRL (Wang et al., 2023a) algorithm—in both of these cases, however, we observe negative results.

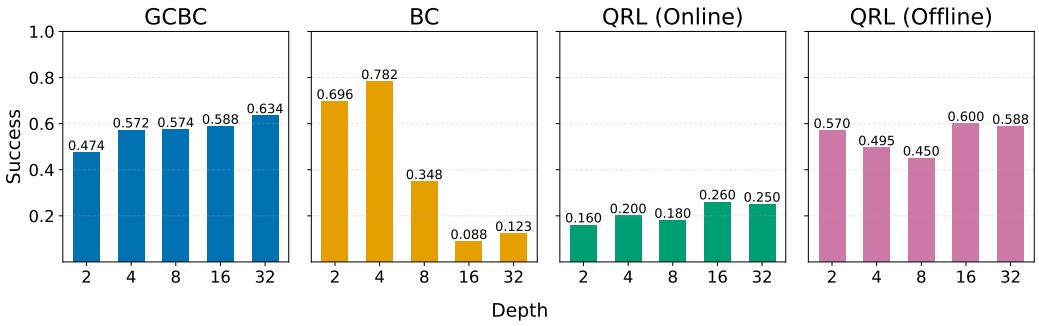

Figure 14: Our approach successfully scales depth in offline GCBC on *antmaze-medium-stitch* (OGBench). In contrast, scaling depth for BC (*antmaze-giant-navigate*, expert SAC data) and for both online (*FetchPush*) and offline QRL (*pointmaze-giant-stitch*, OGBench) yield negative results.

## A.4 Can Depth Scaling also be Effective for Quasimetric Architectures?

Prior work (Wang et al., 2023b; Liu et al., 2023) has found that temporal distances satisfy an important invariance property, suggesting the use of quasimetric architectures when learning temporal distances. Our next experiment tests whether changing the architecture affects the scaling properties of self-supervised RL. Specifically, we use the CMD-1 algorithm (Myers et al., 2024), which employs a backward NCE loss with MRN representations. The results indicate that scaling benefits are not limited to a single neural network parametrization. However, MRN's poor performance on the Ant U5-Maze task suggests further innovation is needed for consistent scaling with quasimetric models.

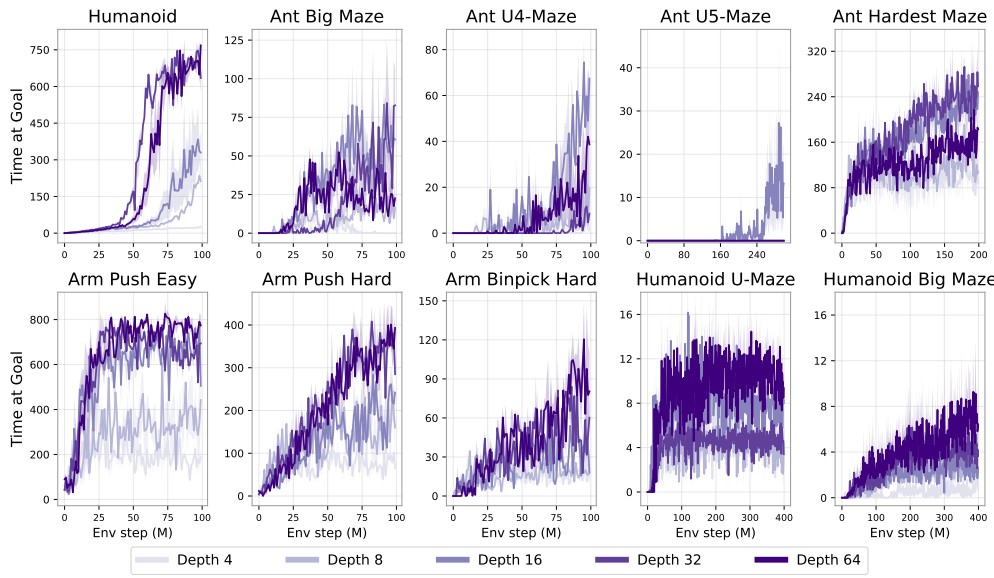

Figure 15: Performance of depth scaling on CRL augmented with quasimetric architectures (CMD-1).

## A.5 Additional Architectural Ablations: Layer Norm and Swish Activation

We conduct ablation experiments to validate the architectural choices of layer norm and swish activation. Figure 16 shows that removing layer normalization performs significantly worse. Additionally, scaling with ReLU significantly hampers scalability. These results, along with Figure 5 show that all of our architectural components—residual connections, layer norm, and swish activations—are jointly essential to unlocking the full performance of depth scaling.

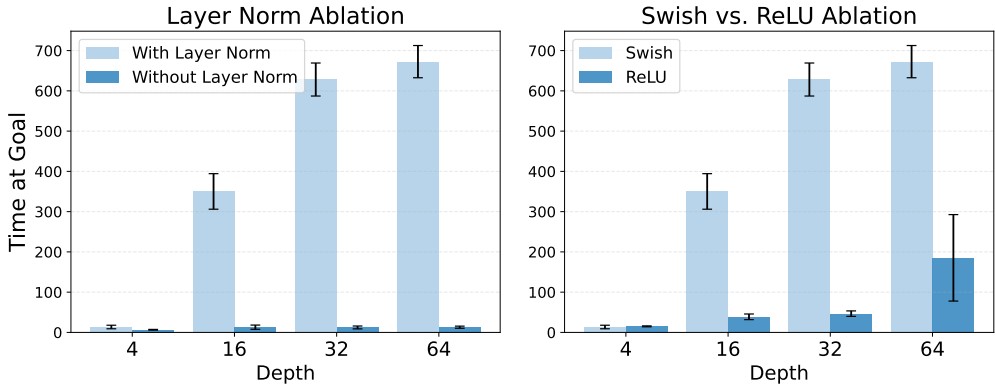

Figure 16: (Left) Layer Norm is essential for scaling depth. (Right) Scaling with ReLU activations leads to worse performance compared to Swish activations.

## A.6 Can We Integrate Novel Architectural Innovations from the Emerging RL Scaling Literature?

Recently, Simba-v2 proposed a new architecture for scalable RL. Its key innovation is the replacement of layer normalization with hyperspherical normalization, which projects network weights onto the unit-norm hypersphere after each gradient update. As shown, the same depth-scaling trends hold when adding hyperspherical normalization to our architecture, and it further improves the sample efficiency of depth scaling. This demonstrates that our method can naturally incorporate new architectural innovations emerging in the RL scaling literature.

Table 2: Integrating hyperspherical normalization in our architecture enhances the sample efficiency of depth scaling.

| Steps to reach ≥200 success | | | | Steps to reach ≥400 success | | | | Steps to reach ≥600 success | | | |
|---|---|---|---|---|---|---|---|---|---|---|---|
| **Depth** | 4 | 16 | 32 | **Depth** | 4 | 16 | 32 | **Depth** | 4 | 16 | 32 |
| **With** | – | **50** | **42** | **With** | – | **62** | **48** | **With** | – | **77** | **67** |
| **Without** | – | 64 | 54 | **Without** | – | 75 | 64 | **Without** | – | – | 77 |

## A.7 Residuals Norms in Deep Networks

Prior work has noted decreasing residual activation norms in deeper layers (Chang et al., 2018). We investigate whether this pattern also holds in our setting. For the critic, the trend is generally evident, especially in very deep architectures (e.g., depth 256). The effect is not as pronounced in the actor.

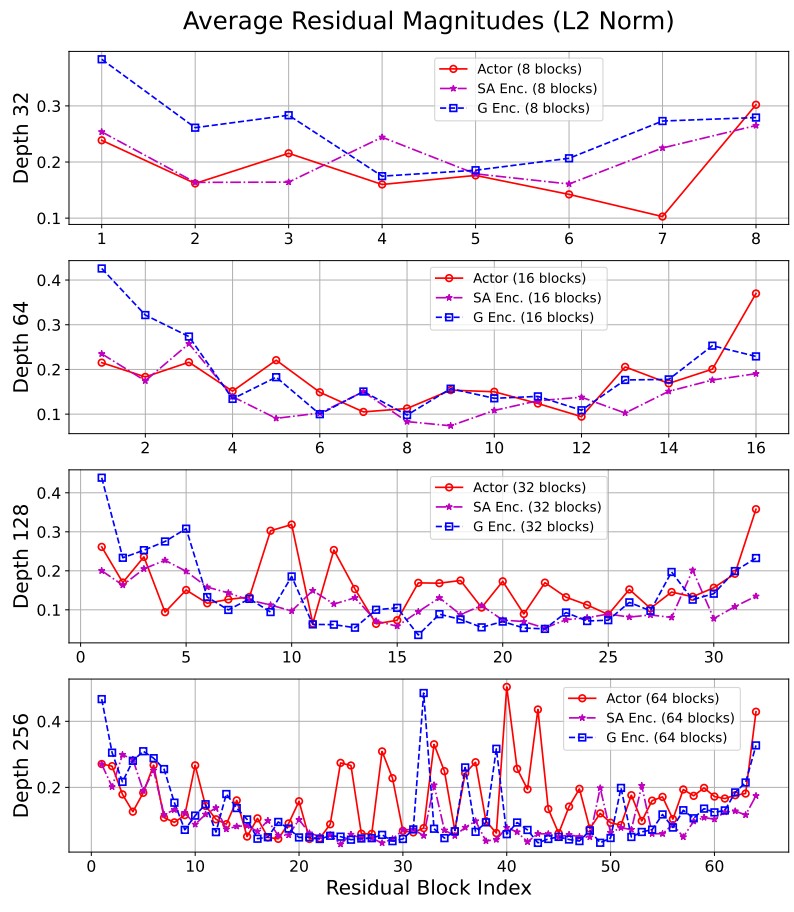

Figure 17: L2 norms of residual activations in networks with depths of 32, 64, 128, and 256.

## A.8 Scaling Depth for Offline Goal-conditioned RL

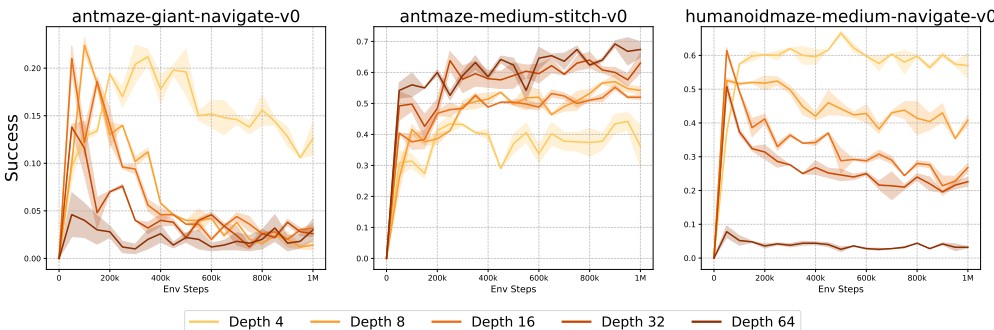

Figure 18: To evaluate the scalability of our method in the offline setting, we scaled model depth on OGBench (Park et al., 2024). In two out of three environments, performance drastically declined as depth scaled from 4 to 64, while a slight improvement was seen on antmaze-medium-stitch-v0. Successfully adapting our method to scale offline GCRL is an important direction for future work.

# B Experimental Details

## B.1 Environment Setup and Hyperparameters

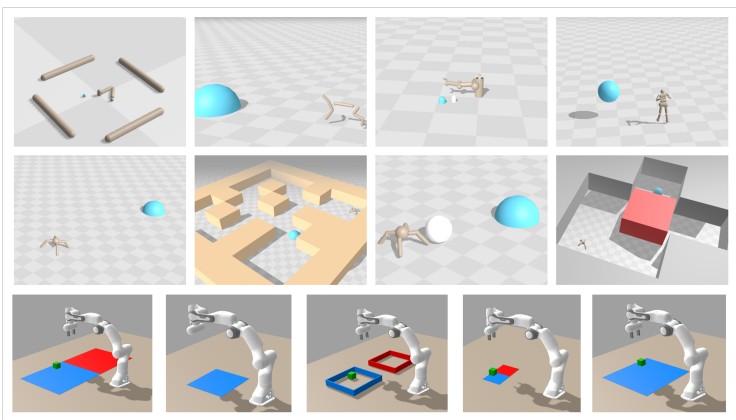

Figure 19: The scaling results of this paper are demonstrated on the JaxGCRL benchmark, showing that they replicate across a diverse range of locomotion, navigation, and manipulation tasks. These tasks are set in the online goal-conditioned setting where there are no auxiliary rewards or demonstrations. Figure taken from (Bortkiewicz et al., 2024).

Our experiments use the JaxGCRL suite of GPU-accelerated environments, visualized in Figure 19, and a contrastive RL algorithm with hyperparameters reported in Table 7. In particular, we use 10 environments, namely: `ant_big_maze`, `ant_hardest_maze`, `arm_binpick_hard`, `arm_push_easy`, `arm_push_hard`, `humanoid`, `humanoid_big_maze`, `humanoid_u_maze`, `ant_u4_maze`, `ant_u5_maze`.

## B.2 Python Environment Differences

In all plots presented in the paper, we used MJX 3.2.6 and Brax 0.10.1 to ensure a fair and consistent comparison. During development, we noticed discrepancies in physics behavior between the environment versions we employed (the CleanRL version of JaxGCRL) and the version recommended in a more recent commit of JaxGCRL (Bortkiewicz et al., 2024). Upon examination, the performance differences (shown in Figure 20) stem from a difference in versions in the MJX and Brax packages. Nonetheless, in both sets of MJX and Brax versions, performance scales monotonically with depth.

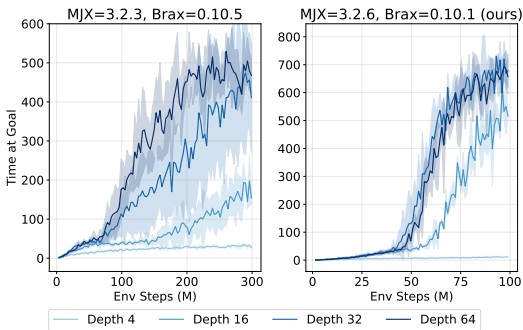

Figure 20: Scaling behavior for humanoid in two different python environments: MJX=3.2.3, Brax=0.10.5 and MJX=3.2.6, Brax=0.10.1 (ours) version of JaxGCRL. Scaling depth improves the performance significantly for both versions. In the environment we used, training requires fewer environment steps to reach a marginally better performance than in other Python environment.

## B.3 Wall-clock Time of Our Approach

We report the wall-clock time of our approach in Table 3. The table shows results for depths of 4, 8, 16, 32, and 64 across all ten environments, and for the Humanoid U-Maze environment, scaling up to 1024 layers. Overall, wall-clock time increases approximately linearly with depth beyond a certain point.

Table 3: Wall-clock time (in hours) for Depth 4, 8, 16, 32, and 64 across all 10 environments.

| Environment | Depth 4 | Depth 8 | Depth 16 | Depth 32 | Depth 64 |
|---|---|---|---|---|---|
| Humanoid | $1.48 \pm 0.00$ | $2.13 \pm 0.01$ | $3.40 \pm 0.01$ | $5.92 \pm 0.01$ | $10.99 \pm 0.01$ |
| Ant Big Maze | $2.12 \pm 0.00$ | $2.77 \pm 0.00$ | $4.04 \pm 0.01$ | $6.57 \pm 0.02$ | $11.66 \pm 0.03$ |
| Ant U4-Maze | $1.98 \pm 0.27$ | $2.54 \pm 0.01$ | $3.81 \pm 0.01$ | $6.35 \pm 0.01$ | $11.43 \pm 0.03$ |
| Ant U5-Maze | $9.46 \pm 1.75$ | $10.99 \pm 0.02$ | $16.09 \pm 0.01$ | $31.49 \pm 0.34$ | $46.40 \pm 0.12$ |
| Ant Hardest Maze | $5.11 \pm 0.00$ | $6.39 \pm 0.00$ | $8.94 \pm 0.01$ | $13.97 \pm 0.01$ | $23.96 \pm 0.06$ |
| Arm Push Easy | $9.97 \pm 1.03$ | $11.02 \pm 1.29$ | $12.20 \pm 1.43$ | $14.94 \pm 1.96$ | $19.52 \pm 1.97$ |
| Arm Push Hard | $9.74 \pm 1.05$ | $10.55 \pm 1.20$ | $11.98 \pm 1.49$ | $14.40 \pm 1.64$ | $18.53 \pm 0.06$ |
| Arm Binpick Hard | $18.41 \pm 2.16$ | $17.48 \pm 1.88$ | $19.47 \pm 0.05$ | $21.91 \pm 1.93$ | $29.64 \pm 6.10$ |
| Humanoid U-Maze | $8.72 \pm 0.01$ | $11.29 \pm 0.01$ | $16.36 \pm 0.03$ | $26.48 \pm 0.05$ | $46.74 \pm 0.04$ |
| Humanoid Big Maze | $12.45 \pm 0.02$ | $15.02 \pm 0.01$ | $20.34 \pm 0.01$ | $30.61 \pm 0.05$ | $50.33 \pm 0.05$ |

Table 4: Total wall-clock time (in hours) for training from Depth 4 up to Depth 1024 in the Humanoid U-Maze environment.

| Depth | Time (h) |
|---|---|
| 4 | $3.23 \pm 0.001$ |
| 8 | $4.19 \pm 0.003$ |
| 16 | $6.07 \pm 0.003$ |
| 32 | $9.83 \pm 0.006$ |
| 64 | $17.33 \pm 0.003$ |
| 128 | $32.67 \pm 0.124$ |
| 256 | $73.83 \pm 2.364$ |
| 512 | $120.88 \pm 2.177$ |
| 1024 | $134.15 \pm 0.081$ |

### B.4 Wall-clock Time: Comparison to Baselines

Since the baselines use standard sized networks, naturally our scaled approach incurs higher raw wall-clock time per environment step (Table 5). However, a more practical metric is the time required to reach a given performance level. As shown in Table 6, our approach outperforms the strongest baseline, SAC, in 7 of 10 environments while requiring less wall-clock time.

Table 5: Wall-clock training time comparison of our method vs. baselines across all 10 environments.

| Environment | Scaled CRL | SAC | SAC+HER | TD3 | GCSL | GCBC |
|---|---|---|---|---|---|---|
| Humanoid | $11.0 \pm 0.0$ | $0.5 \pm 0.0$ | $0.6 \pm 0.0$ | $0.8 \pm 0.0$ | $0.4 \pm 0.0$ | $0.6 \pm 0.0$ |
| Ant Big Maze | $11.7 \pm 0.0$ | $1.6 \pm 0.0$ | $1.6 \pm 0.0$ | $1.7 \pm 0.0$ | $1.5 \pm 0.3$ | $1.4 \pm 0.1$ |
| Ant U4-Maze | $11.4 \pm 0.0$ | $1.2 \pm 0.0$ | $1.3 \pm 0.0$ | $1.3 \pm 0.0$ | $0.7 \pm 0.0$ | $1.1 \pm 0.1$ |
| Ant U5-Maze | $46.4 \pm 0.1$ | $5.7 \pm 0.0$ | $6.1 \pm 0.0$ | $6.2 \pm 0.0$ | $2.8 \pm 0.1$ | $5.6 \pm 0.5$ |
| Ant Hardest Maze | $24.0 \pm 0.0$ | $4.3 \pm 0.0$ | $4.5 \pm 0.0$ | $5.0 \pm 0.0$ | $2.1 \pm 0.6$ | $4.4 \pm 0.5$ |
| Arm Push Easy | $19.5 \pm 0.6$ | $8.3 \pm 0.0$ | $8.5 \pm 0.0$ | $8.4 \pm 0.0$ | $6.4 \pm 0.1$ | $8.3 \pm 0.3$ |
| Arm Push Hard | $18.5 \pm 0.0$ | $8.5 \pm 0.0$ | $8.6 \pm 0.0$ | $8.3 \pm 0.1$ | $5.2 \pm 0.3$ | $7.4 \pm 0.5$ |
| Arm Binpick Hard | $29.6 \pm 1.3$ | $20.7 \pm 0.1$ | $20.7 \pm 0.0$ | $18.4 \pm 0.3$ | $8.0 \pm 0.9$ | $16.2 \pm 0.4$ |
| Humanoid U-Maze | $46.7 \pm 0.0$ | $3.0 \pm 0.0$ | $3.5 \pm 0.0$ | $5.4 \pm 0.0$ | $3.1 \pm 0.1$ | $7.2 \pm 0.8$ |
| Humanoid Big Maze | $50.3 \pm 0.0$ | $8.6 \pm 0.0$ | $9.3 \pm 0.0$ | $7.5 \pm 1.1$ | $5.1 \pm 0.0$ | $11.4 \pm 1.9$ |

Table 6: Wall-clock time (in hours) for our approach to surpass SAC's final performance. As shown, our approach surpasses SAC performance in less wall-clock time in 7 out of 10 environments. The N/A* entries are because in those environments, scaled CRL doesn't outperform SAC.

| Environment | SAC | Scaled CRL (Depth 64) |
|---|---|---|
| Humanoid | **0.46** | 6.37 |
| Ant Big Maze | 1.55 | **0.00** |
| Ant U4-Maze | 1.16 | **0.00** |
| Ant U5-Maze | 5.73 | **0.00** |
| Ant Hardest Maze | 4.33 | **0.45** |
| Arm Push Easy | 8.32 | **1.91** |
| Arm Push Hard | 8.50 | **6.65** |
| Arm Binpick Hard | 20.70 | **4.43** |
| Humanoid U-Maze | **3.04** | N/A* |
| Humanoid Big Maze | **8.55** | N/A* |

Table 7: Hyperparameters

| Hyperparameter | Value |
|---|---|
| num_timesteps | 100M-400M (varying across tasks) |
| update-to-data (UTD) ratio | 1:40 |
| max_replay_size | 10,000 |
| min_replay_size | 1,000 |
| episode_length | 1,000 |
| discounting | 0.99 |
| num_envs | 512 |
| batch_size | 512 |
| policy_lr | 3e-4 |
| critic_lr | 3e-4 |
| contrastive_loss_function | InfoNCE |
| energy_function | L2 |
| logsumexp_penalty | 0.1 |
| Network depth | depends on the experiment |
| Network width | depends on the experiment |
| representation dimension | 64 |

