# OpenReview forum: "1000 Layer Networks for Self-Supervised RL: Scaling Depth Can Enable New Goal-Reaching Capabilities"
_NeurIPS.cc/2025/Conference — NeurIPS 2025 oral_

### Official Review · Reviewer_4F8D · 2025-06-20

**Clarity:** 4
**Significance:** 3
**Originality:** 2
**Rating:** 5
**Confidence:** 5

**Summary:**

This paper explores the potential of deep network architectures in self-supervised reinforcement learning (RL) to enhance scalability. Unlike most recent RL studies that use shallow networks (2-5 layers), the authors show that increasing network depth to up to 1024 layers can significantly improve performance. The experiments are conducted in an unsupervised, goal-conditioned setting without demonstrations or rewards, requiring the agent to explore and learn to reach commanded goals. Tested on simulated locomotion and manipulation tasks, the approach achieves 2x-50x performance improvements over other goal-conditioned baselines, demonstrating that greater depth not only boosts success rates but also qualitatively alters the learned behaviors.

**Questions:**

1. How do the training costs of the proposed algorithm compare with those of the baselines? Can anthors show it in a Table？
2. Can the authors elaborate empirically or conceptually on why performance “jumps” occur at specific depths or why agents exhibit different behaviors at certain depths (as shown in Figure 3) in certain environments?

**Ethical Concerns:**

["NO or VERY MINOR ethics concerns only"]

**Final Justification:**

The authors' response has addressed my concerns, and I will maintain my score.

**Limitations:**

yes

**Paper Formatting Concerns:**

There are no formatting issues in this paper.

**Quality:**

3

**Strengths And Weaknesses:**

**Strengths**

1. The idea is interesting, and the writing is clear and easy to understand.
2. The authors conducted numerous experiments to demonstrate that increasing the depth of networks is a key factor in improving the performance of CRL.
3. The authors carried out extensive analysis experiments to explore various issues, including some that the reviewer might be concerned about. For example, whether increasing the depth of networks can enhance the performance of other baseline algorithms. Although the answer is no, combining scaling depth or network size with other algorithms may be one of the important future directions for RL.

**Weaknesses**

1. There is an error in the citation of the appendix at line 173, which shows as "？？".
2. The paper does not mention the training and inference costs of the proposed algorithm. The training and inference costs associated with a network depth of 1000 layers may be relatively high.
3. The innovativeness of the article has certain deficiencies. The core technologies involved (such as residual connections, normalization, activation functions, etc.) are derived from existing research. The main novelty of the article lies in the impact of the increase in model size on performance, rather than innovation in methodology.
4. The contributions of the article are mainly experimental, with a lack of theoretical discussion on the critical parts of the experiments. For example, why does the performance undergo a sudden change when the depth reaches a certain critical point? This, to some extent, limits the article's insight.

---

> ### Author Rebuttal · Authors · 2025-07-31
>
> We thank Reviewer 4F8D for the helpful feedback, questions, and suggestions.
>
> > **1. How do the training costs of the proposed algorithm compare with those of the baselines? Can authors show it in a table?**
>
> Sure! Because the baseline algorithms use standard-sized networks (less than 4 layers), our scaled approach naturally incurs higher raw wall-clock time per environment step (**Table 1a**). However, for practical deployment the more meaningful metric is the total wall-clock time required to attain a given level of performance. To evaluate this, we compare our method with the strongest baseline, SAC. As shown in **Table 1b**, our approach surpasses SAC performance in less wall-clock time in 7 out of 10 environments.
>
> > **2. Can the authors elaborate empirically or conceptually on why performance “jumps” occur at specific depths or why agents exhibit different behaviors at certain depths (as shown in Figure 3) in certain environments?**
>
> To elaborate conceptually why performance “jumps” occur at specific depths, we draw parallels to the broader deep learning literature of the well-documented phenomenon of *emergent abilities*, where a sudden increase in performance arises at a critical threshold of model scale. Wei et al. 2022 [1] first proposed that large language models exhibit such emergent behavior, with performance transitioning sharply from near-random at a certain threshold points in model size. Although cross-entropy loss improves smoothly with scale, the model only begins to complete tasks once a critical threshold of capability is reached. In our experiments, we observe similar patterns in the Humanoid agent: as the depth increases from 4 to 8 to 16 layers, its ability to locomote improves gradually. However, only upon reaching a critical level of coordination—when it can walk stably without falling—does the agent achieve the goal and remain upright. This manifests as a sharp, discontinuous jump in performance. Wei et al. also remark that emergent abilities often involve qualitatively novel behaviors, such as chain-of-thought reasoning in LLMs, that are absent in smaller models. We observe a parallel in the Humanoid U-Maze environment: only at a very large model scale (i.e., depth 256) does the agent achieve the complex and coordinated movement necessary to successfully vault over the wall. At smaller depths, the agent simply appears to “flounder”, its movements not yet sophisticated enough to complete the maneuver. This behavior manifests as a qualitatively distinct, “emergent” ability and is reflected in the observed performance jump.
>
>
> > **3. There is an error in the citation of the appendix at line 173, which shows as "？？".**
>
> Thank you for pointing that out. This figure comparing our approach with baselines is included as Figure 10 of the Appendix. The issue was due to a typo that resulted in an undefined reference — we have now fixed this in our local version.
>
> > **4. The paper does not mention the training and inference costs of the proposed algorithm. The training and inference costs associated with a network depth of 1000 layers may be relatively high.**
>
> Thanks for this suggestion. In terms of considerations on training and inference costs, we report the wall-clock time of our approach as we scale depth (**Table 2a**, **Table 2b**). A comparison of the cost of wall-clock time vs. baselines is discussed in the first paragraph of this response above, and a comparison of wall-clock time for depth vs. width scaling can be found in **Table 3a** and **Table 3b**. We likewise also report the metrics of FLOPs and memory (**Table 4**, **Table 5**).
>
> > **5. For example, why does the performance undergo a sudden change when the depth reaches a certain critical point? This, to some extent, limits the article's insight.**
>
> We address this in the second paragraph of this response above.
>
> ### [TABLE 1a] Wall-clock Time Comparison of Our Method vs. Baselines
> |Method|Humanoid|Ant Big Maze|Ant U4-Maze|Ant U5-Maze|Ant Hardest Maze|Arm Push Easy|Arm Push Hard|Arm Binpick Hard|Humanoid U-Maze|Humanoid Big Maze|
> |---|---|---|---|---|---|---|---|---|---|---|
> |CRL Depth 4|1.48 ± 0.00|2.12 ± 0.00|1.98 ± 0.05|9.46 ± 0.28|5.11 ± 0.00|9.97 ± 0.39|9.74 ± 0.40|18.41 ± 0.82|8.72 ± 0.00|12.45 ± 0.00|
> |CRL Depth 32|5.92 ± 0.00|6.57 ± 0.01|6.35 ± 0.00|31.49 ± 0.13|13.97 ± 0.00|14.94 ± 0.88|14.40 ± 0.62|21.91 ± 0.38|26.48 ± 0.02|30.61 ± 0.02|
> |CRL Depth 64|10.99 ± 0.00|11.66 ± 0.01|11.43 ± 0.01|46.40 ± 0.05|23.96 ± 0.03|19.52 ± 0.57|18.53 ± 0.02|29.64 ± 1.30|46.74 ± 0.02|50.33 ± 0.02|
> |SAC|0.46 ± 0.00|1.55 ± 0.00|1.16 ± 0.00|5.73 ± 0.01|4.33 ± 0.00|8.32 ± 0.00|8.49 ± 0.04|20.70 ± 0.07|3.04 ± 0.00|8.55 ± 0.01|
> |SAC+HER|0.55 ± 0.00|1.63 ± 0.00|1.26 ± 0.00|6.10 ± 0.00|4.51 ± 0.00|8.54 ± 0.03|8.60 ± 0.01|20.74 ± 0.02|3.48 ± 0.00|9.27 ± 0.01|
> |TD3|0.79 ± 0.01|1.68 ± 0.01|1.30 ± 0.00|6.18 ± 0.01|5.04 ± 0.00|8.41 ± 0.01|8.31 ± 0.07|18.40 ± 0.28|5.35 ± 0.00|7.50 ± 1.14|
> |GCSL|0.42 ± 0.01|1.48 ± 0.26|0.70 ± 0.00|2.81 ± 0.12|2.13 ± 0.57|6.37 ± 0.12|5.22 ± 0.34|7.95 ± 0.90|3.07 ± 0.08|5.08 ± 0.04|
> |GCBC|0.61 ± 0.03|1.40 ± 0.13|1.11 ± 0.08|5.56 ± 0.54|4.38 ± 0.52|8.28 ± 0.34|7.36 ± 0.50|16.24 ± 0.36|7.18 ± 0.78|11.37 ± 1.93|
>
> ### [TABLE 1b] CRL Surpasses SAC Performance in Less Wall-clock Time in 7 of 10 Environments
> | Algorithm                 |     Humanoid |  Ant Big Maze |   Ant U4-Maze |   Ant U5-Maze | Ant Hardest Maze |   Arm Push Easy |    Arm Push Hard | Arm Binpick Hard | Humanoid U-Maze | Humanoid Big Maze |
> | ------------------------- | -----------: | ------------: | ------------: | ------------: | ---------------: | --------------: | ---------------: | ---------------: | --------------: | ----------------: |
> | **SAC**                   |         **27.8** |          92.7 |          69.8 |         343.8 |            259.8 |           499.2 |            509.7 |           1242.1 |           **182.6** |             **512.9** |
> | **Scaled CRL (Depth 64)** | 382.2 ± 11.3 | **0.0 ± 0.0** | **0.0 ± 0.0** | **0.0 ± 0.0** |   **27.2 ± 3.8** | **114.7 ± 9.9** | **398.8 ± 42.1** | **266.0 ± 14.7** |           N/A\* |             N/A\* |
>
> *N/A denotes SAC outperforms our approach, so CRL never reaches SAC performance. These account for 2 of the 3 environments where CRL fails to surpass SAC in less wall-clock time.
>
>
> ### [Table 2a] Wallclock time of Depth 4, 8, 16, 32, and 64 across all 10 environments
> |Depth|Humanoid|Ant Big Maze|Ant U4-Maze|Ant U5-Maze|Ant Hardest Maze|Arm Push Easy|Arm Push Hard|Arm Binpick Hard|Humanoid U-Maze|Humanoid Big Maze|
> |---|---|---|---|---|---|---|---|---|---|---|
> |Depth 4|1.48 ± 0.00|2.12 ± 0.00|1.98 ± 0.27|9.46 ± 1.75|5.11 ± 0.00|9.97 ± 1.03|9.74 ± 1.05|18.41 ± 2.16|8.72 ± 0.01|12.45 ± 0.02|
> |Depth 8|2.13 ± 0.01|2.77 ± 0.00|2.54 ± 0.01|10.99 ± 0.02|6.39 ± 0.00|11.02 ± 1.29|10.55 ± 1.20|17.48 ± 1.88|11.29 ± 0.01|15.02 ± 0.01|
> |Depth 16|3.40 ± 0.01|4.04 ± 0.01|3.81 ± 0.01|16.09 ± 0.01|8.94 ± 0.01|12.20 ± 1.43|11.98 ± 1.49|19.47 ± 0.05|16.36 ± 0.03|20.34 ± 0.01|
> |Depth 32|5.92 ± 0.01|6.57 ± 0.02|6.35 ± 0.01|31.49 ± 0.34|13.97 ± 0.01|14.94 ± 1.96|14.40 ± 1.64|21.91 ± 1.93|26.48 ± 0.05|30.61 ± 0.05|
> |Depth 64|10.99 ± 0.01|11.66 ± 0.03|11.43 ± 0.03|46.40 ± 0.12|23.96 ± 0.06|19.52 ± 1.97|18.53 ± 0.06|29.64 ± 6.10|46.74 ± 0.04|50.33 ± 0.05|
>
> ### [Table 2b] Wallclock time up to Depth 1024 in the Humanoid U-Maze environment
> | Depth        | 4            | 8            | 16           | 32           | 64            | 128           | 256           | 512            | 1024           |
> | ------------ | ------------ | ------------ | ------------ | ------------ | ------------- | ------------- | ------------- | -------------- | -------------- |
> | Time (hours) | 3.23 ± 0.001 | 4.19 ± 0.003 | 6.07 ± 0.003 | 9.83 ± 0.006 | 17.33 ± 0.003 | 32.67 ± 0.124 | 73.83 ± 2.364 | 120.88 ± 2.177 | 134.15 ± 0.081 |
>
> ### [TABLE 3a] Wall-clock Time Comparison of Depth vs. Width Scaling
> |Width 256|Width 512|Width 1024|Width 2048|Width 4096|
> |---|---|---|---|---|
> |4.52 ± 0.34|5.32 ± 0.01|5.96 ± 0.01|8.39 ± 0.02|15.90 ± 0.64|
>
> |Depth 4|Depth 8|Depth 16|Depth 32|Depth 64|
> |---|---|---|---|---|
> |4.52 ± 0.34|5.31 ± 0.43|6.55 ± 0.48|9.14 ± 0.65|14.06 ± 0.66|
>
>
> ### [TABLE 3b] Scaling Depth Surpasses Performance of Scaling Width in Less Wall-clock time
> |Environment|Width = 4096|Depth = 64|
> |-|-|-|
> |Humanoid|22.30 ± 0.08|**5.66 ± 0.24**|
> |Ant Big Maze|11.70 ± 0.03|**4.04 ± 0.41**|
> |Arm Push Easy|13.69 ± 1.92|**2.38 ± 0.18**|
>
>
> ### [TABLE 4] Comparison of Width vs. Depth Scaling: FLOPS and Memory
> FLOPs comparison of width vs. depth scaling
> |Width|256|512|1024|2048|4096|
> |---|---|---|---|---|---|
> |FLOPs (B)|4.8|17.4|66.9|262.4|1039.9|
>
> |Depth|4|8|16|32|64|
> |---|---|---|---|---|---|
> |FLOPs (B)|4.8|8.9|17.1|33.5|66.2|
>
> ### [TABLE 4] Comparison of Width vs. Depth Scaling: FLOPs
> |Width|256|512|1024|2048|4096|
> |---|---|---|---|---|---|
> |FLOPs (B)|4.8|17.4|66.9|262.4|1039.9|
>
> |Depth|4|8|16|32|64|
> |---|---|---|---|---|---|
> |FLOPs (B)|4.8|8.9|17.1|33.5|66.2|
>
> ### [TABLE 5] Comparison of Width vs. Depth Scaling: Memory
> |Width|256|512|1024|2048|4096|
> |---|---|---|---|---|---|
> |Memory (MB)|24.8|63.4|194.7|673.3|2494.6|
>
> |Depth|4|8|16|32|64|
> |---|---|---|---|---|---|
> |Memory (MB)|24.8|43.8|81.9|158.0|310.3|
>
>
> [1] Wei, J., et al., NeurIPS 2022, Chain-of-Thought Prompting Elicits Reasoning in Large Language Models https://arxiv.org/abs/2201.11903

---

> > ### Comment · Reviewer_4F8D · 2025-08-01
> > **Reply to 'Rebuttal by Authors'**
> >
> > The authors' response has addressed my concerns, and I will maintain my score.

---

### Official Review · Reviewer_kTrp · 2025-06-27

**Clarity:** 4
**Significance:** 2
**Originality:** 2
**Rating:** 5
**Confidence:** 4

**Summary:**

The paper investigates scaling the depth of the networks used in self-supervised reinforcement learning for goal-reaching tasks. The work shows that scaling depth can significantly boost the performance of Contrastive RL in the JaxGCRL suite of online RL tasks. The authors also extensively analyze their design choices and provide qualitative analyses of the behavior and the value functions learned.

**Questions:**

- How is training time affected as you scale depth?

Also, I leave here some questions/idea for the authors to test outside of their contrastive RL+ online RL settings:
* Given that simply scaling depth in the actor seems beneficial, would scaling depth be useful for BC and GCBC in Offline RL settings?
* I understand the authors provide a study on CMD-1 in the Appendix, which uses the MRN network for quasimetric learning. It would be useful to also study QRL [1], which instead uses IQE for quasimetric learning

Minor note:
* the reference at Line 173 is undefined (probably due to the Figure being in the Appendix)

[1] Optimal Goal-Reaching Reinforcement Learning via Quasimetric Learning

**Ethical Concerns:**

["NO or VERY MINOR ethics concerns only"]

**Final Justification:**

The work presents depth scaling as a successful technique to scale (self-supervised) RL. The findings seem to be mostly limited to the CRL approach (and to GCBC, as shown in some experiments posted for the rebuttal). However, the work remains an interesting case study and the authors provide an insightful analysis, which I think will be useful to the community.

**Limitations:**

The authors have adequately addressed limitations in their work.

**Paper Formatting Concerns:**

No paper formatting concerns.

**Quality:**

4

**Strengths And Weaknesses:**

## Strengths
* **Strong empirical analysis**: the empirical analysis of the approach is extensive and I found it particularly useful. It clearly shows how scaling depth can significantly improve performance for Contrastive RL. However, it also provides analyses about: scaling width vs depth, the usefulness of residual layers, scaling actor vs critic network, etc.
* **New SOTA on JaxGCRL**: the results obtained with scaling depth and data are impressive. Also, improving the state-of-the-art in a benchmark is useful to further push the boundaries of research on that benchmark and in the field.
## Weaknesses
* **Limited generality of the claims** : my main concern with this work is that the idea of scaling depth seem to apply to a very specific algorithm, i.e. contrastive RL, and in very specific settings, i.e. online RL. I appreciate the authors' attempts in testing their method in offline RL settings and using CMD-1 (in Appendix). However, the generality of their claims remain so far limited.

---

> ### Author Rebuttal · Authors · 2025-07-31
>
> We thank Reviewer kTrp for the helpful feedback, questions, and suggestions.
>
> > **1. How is training time affected as you scale depth?**
>
> We report the wall-clock metrics of our approach as we scale depth in **Table 1**. We also conduct a comparison to baselines: since the baselines use standard sized networks, naturally our scaled approach incurs higher raw wall-clock time per environment step (**Table 2a**). For practical deployment, however, the more meaningful metric is often wall-clock time to attain a given level of performance. In **Table 2b**, we compare against the strongest baseline, SAC, and show that in 7 out of 10 environments our approach surpasses SAC performance in less wall-clock time.
>
> > **2. Limited generality of the claims : my main concern with this work is that the idea of scaling depth seem to apply to a very specific algorithm, i.e. contrastive RL, and in very specific settings, i.e. online RL.**
>
> To address this concern, we report a new experimental result: we find that our approach successfully scales depth on a different algorithm (GCBC) in a different setting (offline RL). In **Table 3a**, we find that GCBC successfully scales on the antmaze-medium-stitch task in OGBench [1] when using our architecture of layer normalization, residual connections, and swish activations. In **Table 3b**, we show that using our architecture is critical to enabling scaling in GCBC, as scaling the standard architecture degrades performance. These results suggest that our approach to depth scaling may be adapted to other algorithms, as well as the offline setting. Future work could further investigate what algorithms and architectural components best enable depth scaling in offline RL.
>
> > **3. Given that simply scaling depth in the actor seems beneficial, would scaling depth be useful for BC and GCBC in Offline RL settings?**
>
> Thanks for this suggestion! For GCBC, we have discussed the results in the response above. We also attempt to scale depth for standard BC in OGBench [1]. However, in this case we find negative results (**Table 4**).
>
> > **4. I understand the authors provide a study on CMD-1 in the Appendix, which uses the MRN network for quasimetric learning. It would be useful to also study QRL [1], which instead uses IQE for quasimetric learning**
>
> We attempt to scale depth for QRL in both the online and offline settings, using the original QRL codebase [2] and OGBench [1] respectively. However, in both settings, we find that while QRL is able to train stably with deep networks, deep networks don’t improve performance beyond that of the shallow networks. **Table 5a** and **Table 5b** show these results.
>
> ### [Table 1a] Wallclock time of Depth 4, 8, 16, 32, and 64 across all 10 environments
> |Depth|Humanoid|Ant Big Maze|Ant U4-Maze|Ant U5-Maze|Ant Hardest Maze|Arm Push Easy|Arm Push Hard|Arm Binpick Hard|Humanoid U-Maze|Humanoid Big Maze|
> |---|---|---|---|---|---|---|---|---|---|---|
> |Depth 4|1.48 ± 0.00|2.12 ± 0.00|1.98 ± 0.27|9.46 ± 1.75|5.11 ± 0.00|9.97 ± 1.03|9.74 ± 1.05|18.41 ± 2.16|8.72 ± 0.01|12.45 ± 0.02|
> |Depth 8|2.13 ± 0.01|2.77 ± 0.00|2.54 ± 0.01|10.99 ± 0.02|6.39 ± 0.00|11.02 ± 1.29|10.55 ± 1.20|17.48 ± 1.88|11.29 ± 0.01|15.02 ± 0.01|
> |Depth 16|3.40 ± 0.01|4.04 ± 0.01|3.81 ± 0.01|16.09 ± 0.01|8.94 ± 0.01|12.20 ± 1.43|11.98 ± 1.49|19.47 ± 0.05|16.36 ± 0.03|20.34 ± 0.01|
> |Depth 32|5.92 ± 0.01|6.57 ± 0.02|6.35 ± 0.01|31.49 ± 0.34|13.97 ± 0.01|14.94 ± 1.96|14.40 ± 1.64|21.91 ± 1.93|26.48 ± 0.05|30.61 ± 0.05|
> |Depth 64|10.99 ± 0.01|11.66 ± 0.03|11.43 ± 0.03|46.40 ± 0.12|23.96 ± 0.06|19.52 ± 1.97|18.53 ± 0.06|29.64 ± 6.10|46.74 ± 0.04|50.33 ± 0.05|
>
> ### [Table 1b] Wallclock time up to Depth 1024 in the Humanoid U-Maze environment
> | Depth        | 4            | 8            | 16           | 32           | 64            | 128           | 256           | 512            | 1024           |
> | ------------ | ------------ | ------------ | ------------ | ------------ | ------------- | ------------- | ------------- | -------------- | -------------- |
> | Time  (h) | 3.23 ± 0.001 | 4.19 ± 0.003 | 6.07 ± 0.003 | 9.83 ± 0.006 | 17.33 ± 0.003 | 32.67 ± 0.124 | 73.83 ± 2.364 | 120.88 ± 2.177 | 134.15 ± 0.081 |
>
> ### [TABLE 2a] Wall-clock Time Comparison of Our Method vs. Baselines
> |Method|Humanoid|Ant Big Maze|Ant U4-Maze|Ant U5-Maze|Ant Hardest Maze|Arm Push Easy|Arm Push Hard|Arm Binpick Hard|Humanoid U-Maze|Humanoid Big Maze|
> |---|---|---|---|---|---|---|---|---|---|---|
> |Scaled CRL (Depth 64)|10.99 ± 0.00|11.66 ± 0.01|11.43 ± 0.01|46.40 ± 0.05|23.96 ± 0.03|19.52 ± 0.57|18.53 ± 0.02|29.64 ± 1.30|46.74 ± 0.02|50.33 ± 0.02|
> |SAC|0.46 ± 0.00|1.55 ± 0.00|1.16 ± 0.00|5.73 ± 0.01|4.33 ± 0.00|8.32 ± 0.00|8.49 ± 0.04|20.70 ± 0.07|3.04 ± 0.00|8.55 ± 0.01|
> |SAC+HER|0.55 ± 0.00|1.63 ± 0.00|1.26 ± 0.00|6.10 ± 0.00|4.51 ± 0.00|8.54 ± 0.03|8.60 ± 0.01|20.74 ± 0.02|3.48 ± 0.00|9.27 ± 0.01|
> |TD3|0.79 ± 0.01|1.68 ± 0.01|1.30 ± 0.00|6.18 ± 0.01|5.04 ± 0.00|8.41 ± 0.01|8.31 ± 0.07|18.40 ± 0.28|5.35 ± 0.00|7.50 ± 1.14|
> |GCSL|0.42 ± 0.01|1.48 ± 0.26|0.70 ± 0.00|2.81 ± 0.12|2.13 ± 0.57|6.37 ± 0.12|5.22 ± 0.34|7.95 ± 0.90|3.07 ± 0.08|5.08 ± 0.04|
> |GCBC|0.61 ± 0.03|1.40 ± 0.13|1.11 ± 0.08|5.56 ± 0.54|4.38 ± 0.52|8.28 ± 0.34|7.36 ± 0.50|16.24 ± 0.36|7.18 ± 0.78|11.37 ± 1.93|
>
>
> ### [TABLE 2b] CRL Surpasses SAC Performance in Less Wall-clock Time in 7 of 10 Environments
>
> | Algorithm                 |     Humanoid |  Ant Big Maze |   Ant U4-Maze |   Ant U5-Maze | Ant Hardest Maze |   Arm Push Easy |    Arm Push Hard | Arm Binpick Hard | Humanoid U-Maze | Humanoid Big Maze |
> | ------------------------- | -----------: | ------------: | ------------: | ------------: | ---------------: | --------------: | ---------------: | ---------------: | --------------: | ----------------: |
> | **SAC**                   |         **27.8** |          92.7 |          69.8 |         343.8 |            259.8 |           499.2 |            509.7 |           1242.1 |           **182.6** |             **512.9** |
> | **Scaled CRL (Depth 64)** | 382.2 ± 11.3 | **0.0 ± 0.0** | **0.0 ± 0.0** | **0.0 ± 0.0** |   **27.2 ± 3.8** | **114.7 ± 9.9** | **398.8 ± 42.1** | **266.0 ± 14.7** |           N/A\* |             N/A\* |
>
> *N/A denotes SAC outperforms our approach, so CRL never reaches SAC performance. These account for 2 of the 3 environments where CRL fails to surpass SAC in less wall-clock time.
>
> ### [TABLE 3a] Depth Scaling on GCBC (Antmaze-medium-stitch, OGBench)
> | Depth        | 2            | 4            | 8            | 16           | 32           |
> | ------------ | ------------ | ------------ | ------------ | ------------ | ------------ |
> | Performance | 0.474 ± 0.11 | 0.572 ± 0.04 | 0.574 ± 0.02 | 0.588 ± 0.00 | 0.634 ± 0.05 |
>
> ### [TABLE 3b] Our Architecture is Critical: Depth Scaling on GCBC Fails with Standard Architecture
> | Depth        | 2             | 4             | 8             | 16            | 32            |
> | ------------ | ------------- | ------------- | ------------- | ------------- | ------------- |
> | Performance | 0.532 ± 0.088 | 0.746 ± 0.014 | 0.692 ± 0.024 | 0.698 ± 0.022 | 0.210 ± 0.210 |
>
>
> ### [Table 4] Depth Scaling on BC (Antmaze-giant-navigate on expert SAC policy, OGBench)
> |Depth|2|4|8|16|32|
> |---|---|---|---|---|---|
> |Performance|0.696 ± 0.00|0.782 ± 0.03|0.348 ± 0.00|0.088 ± 0.02|0.123 ± 0.04|
>
>
> ### [Table 5a] Depth Scaling on Online QRL (FetchPush task on original QRL codebase)
> |Depth|2|4|8|16|32|
> |---|---|---|---|---|---|
> |Performance|0.160 ± 0.04|0.200 ± 0.00|0.180 ± 0.00|0.260 ± 0.10|0.250 ± 0.01|
>
>
> ### [Table 5b] Depth Scaling on Offline QRL (Pointmaze-giant-stitch, OGBench)
> |Depth|2|4|8|16|32|
> |---|---|---|---|---|---|
> |Performance|0.570 ± 0.02|0.495 ± 0.12|0.450 ± 0.00|0.600 ± 0.00|0.588 ± 0.00|
>
> [1] Park, S., et al., ICLR 2025, OGBench: Benchmarking Offline Goal-Conditioned RL, https://arxiv.org/abs/2410.20092.
> [2] Optimal Goal-Reaching Reinforcement Learning via Quasimetric Learning

---

> > ### Comment · Reviewer_kTrp · 2025-08-03
> >
> > I thank the authors for providing additional insights about the training times of their experiments and the possibility of applying the proposed technique to other approaches, i.e. GCBC and QRL. I encourage them to add these experiments to the paper (even in Appendix) as I believe they will provide useful insights to future readers too.
> >
> > I will keep my positive judgement of this paper: I think it should be accepted to the conference.

---

> > > ### Author Response · Authors · 2025-08-03
> > >
> > > We sincerely thank the reviewer for their positive assessment of our work and for providing concrete recommendations for additional experiments. As suggested, we will incorporate these experiments and their insights into the camera-ready version of the paper.

---

### Official Review · Reviewer_5GqS · 2025-07-02

**Clarity:** 3
**Significance:** 3
**Originality:** 3
**Rating:** 5
**Confidence:** 4

**Summary:**

The authors show that depth scaling properties arise in contrastive RL, whereas width scaling is prominent in recent vanilla RL literature [1,2]. They propose to stack neural network blocks with residual connection, normalization (LayerNorm), and Swish architecture. By increasing the number of blocks, authors found that goal reaching capabilities emerge after a certain threshold, shown both by increased performance measure (figure 1) and novel behavior (figure 3). In their analysis, they find that depth scaling improves contrastive representations, enhances exploration, and allows trajectory stitching.

[1] SimBa: Simplicity Bias for Scaling Up Parameters in Deep Reinforcement Learning., Lee et al.

[2] Bigger, Regularized, Categorical: High-Capacity Value Functions are Efficient Multi-Task Learners., Nauman et al.

**Questions:**

Please see weaknesses above.

**Ethical Concerns:**

["NO or VERY MINOR ethics concerns only"]

**Final Justification:**

The concerns I have raised have been addressed with additional experiment; I have raised my score from 4 to 5 accordingly. I believe this paper nicely reveals the scaling potential of GCRL compared to standard RL settings.

**Limitations:**

Yes

**Paper Formatting Concerns:**

None.

**Quality:**

3

**Strengths And Weaknesses:**

Strengths
- Strong performance and clear depth-scaling tendency to support their claims.
- In-depth analysis on the architecture and policy as depth scales.
- Verification of contrastive RL algorithm compared to baseline methods.

Weaknesses
- The experiments are limited to one type of architecture. Whether the same trend would occur in other well-designed architectures in RL [1,2,3] is unsure.
- On a similar note, no sufficient analysis was made on the design choices of the architecture, other than the residual connection.
- The connection between depth and exploration is not well explained. The results presented in figure 13 could also be due to improved learning capability of the collector.
- Due to the model's sequential nature, a depth-scaled model is likely to take more time to generate output than a width-scaled model with same number of parameters. The claim that depth scaling is more compute efficient than width scaling (line 225:229) should thus we accompanied by a comparison on wall-clock time.
- The fact that their architecture does not scale in offline GCRL weakens their point.

Overall, I think the paper provides interesting and useful knowledge to the community, although there are several issues that needs to be refined.

[1] SimBa: Simplicity Bias for Scaling Up Parameters in Deep Reinforcement Learning., Lee et al.

[2] Hyperspherical Normalization for Scalable Deep Reinforcement Learning., Lee et al.

[3] Bigger, Regularized, Optimistic: scaling for compute and sample-efficient continuous control., Nauman et al.

---

> ### Author Rebuttal · Authors · 2025-07-31
>
> We thank Reviewer 5GqS for the helpful feedback, questions, and suggestions.
>
> > **1. The experiments are limited to one type of architecture. Whether the same trend would occur in other well-designed architectures in RL [1,2,3] is unsure.**
>
> To address this concern, we have run a new experiment: we find that the same trend occurs when we apply our approach to another well-designed architecture for scaling RL, Simba-v2 [1]. The key innovation in Simba-v2 is replacing layer norm with hyperspherical normalization, which projects network weights onto the unit-norm hypersphere after each gradient update. **Table 1** shows that indeed the same depth scaling trends holds when we add the hyperspherical normalization to our architecture, and, in fact, actually improves the sample efficiency of depth scaling. This suggests that our method is able to integrate new architectural innovations in the emerging RL scaling literature.
>
>
> > **2. On a similar note, no sufficient analysis was made on the design choices of the architecture, other than the residual connection.**
>
> As suggested by the reviewer, we report new ablation experiments for the other design choices of architecture: namely, layer norm and swish activation. We show that both of these design choices are critical for effective scaling:
> - In **Table 2a**, we show that removing layer normalization significantly hampers scalability.
> - Additionally, **Table 2b** shows that scaling with ReLU activations performs significantly worse than Swish activations.
>
> These results, along with Figure 5 of the manuscript, show that all of our architectural components—residual connections, layer norm, and swish activations—are jointly essential to unlocking the full performance of depth scaling.
>
> > **3. The connection between depth and exploration is not well explained. The results presented in figure 13 could also be due to improved learning capability of the collector.**
>
> To clarify, by exploration, we specifically mean the breadth or coverage of states visited. Shallow networks, due to their limited learning capacity, tend to explore only states near their initializations. While deep networks, due to their greater learning capacity, learn effective long-horizon behaviors, resulting empirically in much broader state coverage. Figure 13’s shallow collector experiments illustrate that improved learning capacity alone is insufficient when data diversity or coverage is poor. Conversely, Figure 13’s deep collector experiments show that good data coverage alone, without adequate learning capacity, also falls short. Thus, rather than viewing learning capacity and exploration as diametrically opposed, they mutually reinforce each other: stronger learning capacity drives more extensive exploration, and rich data coverage is essential to fully realize the potential of stronger learning capacity. Both aspects jointly contribute to improved performance.
>
> > **4. Due to the model's sequential nature, a depth-scaled model is likely to take more time to generate output than a width-scaled model with same number of parameters. The claim that depth scaling is more compute efficient than width scaling (line 225:229) should thus we accompanied by a comparison on wall-clock time.**
>
> In **Table 3a**, we report the average wall-clock time for scaled width and depth, and find that width and depth empirically scale quite similarly in terms of wall-clock time. For practical deployment, the relevant metric is often wall-clock time to reach a given level of performance. **Table 3b** shows that scaling depth surpasses the performance of scaling width in less wall-clock time in all three environments.
>
> > **5. The fact that their architecture does not scale in offline GCRL weakens their point.**
>
> Although we found negative results for scaling CRL in the offline setting, we now report new experiments demonstrating that our approach scales effectively to another widely used offline algorithm, GCBC. In **Table 4a**, we find that GCBC successfully scales with depth on the antmaze-medium-stitch task in OGBench [2] when using our architecture of layer normalization, residual connections, and swish activations. In **Table 4b**, we show that using our architecture is critical to enabling this scaling in GCBC, as scaling the standard architecture degrades performance. These results suggest that our approach may be adapted to scaling the offline RL setting. Future work could further investigate what algorithms and architectural components best enable depth scaling in offline RL.
>
> ### [TABLE 1] Hyperspherical Normalization (Simba-v2) Improves Sample Efficiency of Depth Scaling
>
> **Steps to reach ≥ 200 success**
> | Depth | With | Without |
> | ----: | --------------------------------: | -----------------------------------: |
> |     4 |                                 – |                                    – |
> |    16 |                            **50** |                                   64 |
> |    32 |                            **42** |                                   54 |
>
> **Steps to reach ≥ 400 success**
> | Depth | With | Without |
> | ----: | --------------------------------: | -----------------------------------: |
> |     4 |                                 – |                                    – |
> |    16 |                            **62** |                                   75 |
> |    32 |                            **48** |                                   64 |
>
> **Steps to reach ≥ 600 success**
> | Depth | With | Without |
> | ----: | --------------------------------: | -----------------------------------: |
> |     4 |                                 – |                                    – |
> |    16 |                            **77** |                                    – |
> |    32 |                            **67** |                                   77 |
>
>
> ### [TABLE 2a] Layer Norm Ablation
> | Depth | With Layer Norm | Without Layer Norm |
> |-------------|----------------------|----------------------|
> |       4       |     **13.43 ± 4.40**       |    6.58 ± 0.88    |
> |      16       |     **350.17 ± 44.14**     |    13.01 ± 5.23    |
> |      32       |     **628.15 ± 41.00**     |    12.34 ± 3.55    |
> |      64       |     **672.56 ± 40.01**     |    12.94 ± 2.66   |
>
> ### [TABLE 2b] Swish vs. ReLU Ablation
> | Depth | Swish | ReLU |
> |-------------|----------------------|----------------------|
> |       4       |     **13.43 ± 4.40**       |     15.09 ± 1.15     |
> |      16       |     **350.17 ± 44.14**     |     38.76 ± 6.99     |
> |      32       |     **628.15 ± 41.00**     |     46.62 ± 6.81     |
> |      64       |     **672.56 ± 40.01**     |   185.25 ± 107.40    |
>
> ### [TABLE 3a] Wall-clock Time Comparison of Depth vs. Width Scaling
> |Width 256|Width 512|Width 1024|Width 2048|Width 4096|
> |---|---|---|---|---|
> |4.52 ± 0.34|5.32 ± 0.01|5.96 ± 0.01|8.39 ± 0.02|15.90 ± 0.64|
>
> |Depth 4|Depth 8|Depth 16|Depth 32|Depth 64|
> |---|---|---|---|---|
> |4.52 ± 0.34|5.31 ± 0.43|6.55 ± 0.48|9.14 ± 0.65|14.06 ± 0.66|
>
> ### [TABLE 3b] Scaling Depth Surpasses Performance of Scaling Width in Less Wall-clock time
> |Environment|Width = 4096|Depth = 64|
> |-|-|-|
> |Humanoid|22.30 ± 0.08|**5.66 ± 0.24**|
> |Ant Big Maze|11.70 ± 0.03|**4.04 ± 0.41**|
> |Arm Push Easy|13.69 ± 1.92|**2.38 ± 0.18**|
>
>
> ### [TABLE 4a] Successful Depth Scaling on GCBC (Antmaze-medium-stitch, OGBench)
> | Depth        | 2            | 4            | 8            | 16           | 32           |
> | ------------ | ------------ | ------------ | ------------ | ------------ | ------------ |
> | Performance | 0.474 ± 0.11 | 0.572 ± 0.04 | 0.574 ± 0.02 | 0.588 ± 0.00 | 0.634 ± 0.05 |
>
> ### [TABLE 4b] Our Architecture is Critical: Depth Scaling on GCBC Fails with Standard Architecture
> | Depth        | 2             | 4             | 8             | 16            | 32            |
> | ------------ | ------------- | ------------- | ------------- | ------------- | ------------- |
> | Performance | 0.532 ± 0.088 | 0.746 ± 0.014 | 0.692 ± 0.024 | 0.698 ± 0.022 | 0.210 ± 0.210 |
>
> [1] Lee, H., et al., ICML 2025. Hyperspherical Normalization for Scalable Deep Reinforcement Learning. http://arxiv.org/abs/2502.15280.
> [2] Park, S., et al., ICLR 2025, OGBench: Benchmarking Offline Goal-Conditioned RL, https://arxiv.org/abs/2410.20092

---

> > ### Comment · Reviewer_5GqS · 2025-08-02
> >
> > Thank you for the response. I find the new results really interesting and helpful. I will update my score accordingly after the discussion.
> >
> > That said, I am a bit confused about the results on **A4**. The first sentence and Table 3a claim that width and depth have similar scaling tendencies, but Table 3b seems to tell otherwise. Why are the numbers so different between 3a and 3b? What e nvironments were used for 3a?

---

> > > ### Author Response · Authors · 2025-08-02
> > > **Clarification on Width vs. Depth Scaling**
> > >
> > > We appreciate the reviewer finds our results interesting and helpful! Below, we address the reviewer's question by clarifying Table 3a and Table 3b.
> > >
> > > > **The first sentence and Table 3a claim that width and depth have similar scaling tendencies, but Table 3b seems to tell otherwise. Why are the numbers so different between 3a and 3b?**
> > >
> > > To clarify, the first sentence and Table 3a indicate that scaling depth and scaling width have similar effects **on raw wall‑clock time**, i.e. doubling depth increases the wall-clock time similarly as doubling width. This observation is **purely about runtime** and **does not consider model performance**.
> > >
> > > In practice, what we really care about is comparing the wall-clock time **to reach a target level of performance**, which reflects how efficient is each scaling strategy. This is what Table 3b is comparing. Table 3b shows that scaling depth is significantly more efficient than scaling width: it **achieves the same performance in less wall-clock time.**
> > >
> > > > **What environments were used for 3a?**
> > >
> > > We use the same three environments: Humanoid, Ant Big Maze, Arm Push Easy. Table 3a reports the average wall‑clock time across these environments.

---

### Official Review · Reviewer_gFHa · 2025-07-03

**Clarity:** 3
**Significance:** 4
**Originality:** 4
**Rating:** 5
**Confidence:** 3

**Summary:**

This paper investigates scaling network depth in self-supervised reinforcement learning, specifically using the Contrastive RL (CRL) algorithm for goal-conditioned tasks. The authors demonstrate that increasing network depth up to 1024 layers can yield substantial performance improvements in unsupervised goal-reaching tasks. Their approach combines three key components: (1) self-supervised contrastive RL that learns without demonstrations or rewards, (2) GPU-accelerated training frameworks for increased data throughput, and (3) very deep residual networks with architectural stabilization techniques. The experiments are conducted on locomotion, navigation, and manipulation tasks from the JaxGCRL benchmark, showing 2-50× performance improvements over shallow networks typically used in RL (2-5 layers). The paper provides extensive analysis of why depth scaling works, including improved exploration, better representational capacity, and emergent qualitatively different behaviors at critical depth thresholds.

**Questions:**

1. Can you provide any intutive explanation to why the scaling benefits appear specific to CRL?
2. Could you provide detailed computational cost analysis comparing depth vs. width scaling? What are the FLOPS, memory, and wall-clock time trade-offs? How does this inform practical deployment decisions?
3. In Figure 1, the performance of these approaches increases with depth, and then seems to decrease when depth is increased further. What is the reason for such behavior? Is this pattern consistent with further depth experients? If yes, do you have any thoughts on finding the critical depth for particular tasks?
4. In line 173, the figure that compares proposed approach with SAC, SAC+HER, TD3+HER, and CRL seems to be missing?

**Ethical Concerns:**

["NO or VERY MINOR ethics concerns only"]

**Final Justification:**

The authors' response has addressed all my concerns. The authors properly answered all the raised questions, and also provided the required resource information in the form of four tables, which satisfied my initial concerns. Therefore, I will my high judgement score for the paper.

**Limitations:**

Please refer to the weaknesses section.

**Paper Formatting Concerns:**

None.

**Quality:**

3

**Strengths And Weaknesses:**

Strengths:
1. The paper provides a clearly-written comprehensive study showing that depth scaling (up to 1024 layers) can dramatically improve RL performance, contrary to conventional wisdom.
2. The paper shows strong empirical evidence, by demonstrating consistent improvements across diverse tasks with proper statistical reporting (error bars across multiple seeds), across 10 diverse environments spanning locomotion, navigation, and manipulation.
3. I appreciate the thorough analysis through visualization of Q-functions, learned representations, and emergent behaviors (wall vaulting, creative navigation).
4. The approach provides effective adaptation of residual connections, layer normalization, and Swish activation for very deep RL networks.
5. Good set of ablations are provided: width vs. depth, actor vs. critic scaling, batch size effects, and generalization capabilities.

Weaknesses:
1. It seems that the scaling benefits appear specific to CRL. The traditional TD methods (SAC, TD3) show no improvement or even show degradation with depth.
2. The preliminary experiments show no benefits in offline settings, limiting applicability.
3. The experiments are specific to unsupervised RL setting. It is unclear if the benefits may transfer to reward-rich (dense) or imitation learning settings.
4. The paper would benefit from detailed analysis of computational overhead compared to baselines.
5. As acknowledged by the authors, the approach requires substantial computational resources that may limit reproducibility.

---

> ### Author Rebuttal · Authors · 2025-07-31
>
> We thank Reviewer gFHA for the helpful feedback, questions, and suggestions.
>
> > **Question 1: Can you provide any intuitive explanation to why the scaling benefits appear specific to CRL?**
>
> Sure! The key intuitive reason for scaling benefits specific to CRL lies in how it redefines the core learning challenge of RL. Traditional RL faces several well-known scalability challenges: regression to noisy and recursively bootstrapped Q-value targets, gradient instability, plasticity loss, amplifying overestimation biases, etc. In contrast, CRL fundamentally shifts the “heavy lifting” of the RL learning problem from recursively estimating fluctuating Q-values to addressing a simple classification problem: is the sampled future state likely to be along the same trajectory or along a different trajectory as my current state? This transforms the unstable RL problem into a supervised learning classification task based on cross-entropy loss—a setup known to scale exceptionally well in deep learning (and recently also in RL, see Farebrother et al., 2024 [4]). Once CRL scalably learns good contrastive representations through this supervised objective, the remaining learning burden on the actor becomes straightforward: at any given state, simply greedily choose the action that yields the representation most similar to the goal state.
>
> > **Question 2: Could you provide detailed computational cost analysis comparing depth vs. width scaling? What are the FLOPS, memory, and wall-clock time trade-offs? How does this inform practical deployment decisions?**
>
> In **Table 1a**, we show that width and depth empirically scale similarly in terms of wall-clock time. For practical deployment, the relevant metric is wall-clock time to reach a given performance. **Table 1b** shows that scaling depth surpasses the performance of scaling width in less wall-clock time in all three environments. In terms of FLOPs and memory, width scales quadratically while depth scales linearly, making depth more efficient in these metrics as well. Thus, for practical deployment we recommend scaling depth in almost all cases.
>
> > **Question 3: In Figure 1, the performance of these approaches increases with depth, and then seems to decrease when depth is increased further. What is the reason for such behavior? Do you have any thoughts on finding the critical depth for particular tasks?**
>
> Even with techniques like residual connections and layer norm, training very deep networks is known to be tricky—exploding gradients, representation staleness, plasticity loss, etc. This is likely why many RL scaling efforts show results on width scaling but not depth. In our experiments, for tasks where degradation does occur, performance has already scaled to near-optimal. Thus, we hypothesize that additional depth yields diminishing returns while sometimes increasing the risk of instability. In these cases, to answer the question on identifying critical depth:
> - **Start with 32 layers**: Across all 10 environments, we consistently observe stable and improving performance up to depth 32.
> - **Scale deeper if needed**: For more complex tasks, especially those involving locomotion or high-dimensional control (e.g., Humanoid environments), we observe stable scaling up to 1000 layers.
> - **Be conservative with manipulation tasks**: These tasks tend to be more sensitive to over-scaling and performance degradation at higher depths.
>
> > **Question 4: In line 173, the figure seems to be missing?**
>
> This Figure is in the Appendix (Figure 10). Thanks for catching the undefined reference.
>
> > **Weakness 1: It seems that the scaling benefits appear specific to CRL. The traditional TD methods (SAC, TD3) show no improvement or even show degradation with depth.**
>
> This is a known result from prior work that traditional TD methods do not scale well with depth [5, 2, 3]. As such, rather than viewing this as a limitation, we see it as a point of strength/novelty of our work: our results demonstrate that the benefits of scaling depth can indeed be unlocked in RL. Previous efforts to scale RL have largely focused on network width [5,2], with depth scaling either limited (e.g., up to 8 residual blocks in [3]) or harmful [5, 2]. Our approach unlocks the ability to scale depth, yielding performance improvements greater than width alone.
>
> > **Weakness 2: The preliminary experiments show no benefits in offline settings, limiting applicability.**
>
> Although we found negative results for scaling CRL in the offline setting, we now report new experiments demonstrating that our approach scales effectively to another widely used offline algorithm, GCBC. In **Table 2a**, we find that GCBC successfully scales with depth on the antmaze-medium-stitch task in OGBench [1] when using our architecture of layer normalization, residual connections, and swish activations. In **Table 2b**, we show that using our architecture is critical to enabling scaling in GCBC, as scaling the standard architecture degrades performance. These results suggest that our approach may be adapted to scaling the offline RL setting. Future work could further investigate what algorithms and architectural components best enable depth scaling in offline RL.
>
> > **Weakness 3: The experiments are specific to unsupervised RL setting. It is unclear if the benefits may transfer to reward-rich (dense) or imitation learning settings.**
>
> CRL is an unsupervised RL algorithm; as such, we focus on this setting as the primary scope of our paper. One potentially interesting direction for future research: although CRL itself is unsupervised, its self-supervised contrastive representations can be learned on any set of collected trajectories, including those of a reward-based RL algorithm. Prior work has explored integrating such representations via, for example, auxiliary losses or planning [6]. Our analysis of visualized Q-functions and learned representations give confidence that increasingly sophisticated representations can be learned through depth scaling. We encourage future work to investigate whether these scaling benefits can thus be integrated into reward-based RL.
>
> > **Weakness 4: The paper would benefit from detailed analysis of computational overhead compared to baselines.**
>
> Because the baselines use standard-size networks, our scaled approach naturally incurs higher wall-clock time per environment step at a raw computation level (**Table 3a**). However, as with the width-vs-depth analysis, the more meaningful metric is the total wall-clock time required to reach a given performance level. To evaluate this, we compare our method with the strongest baseline, SAC. As shown in **Table 3b**, our approach surpasses SAC performance in less wall-clock time in 7 out of 10 environments.
>
> > **Weakness 5: The approach requires substantial computational resources that may limit reproducibility.**
>
> In fact, all of our experiments, including those with 1000 layer networks, can be run on a single 80GB A100!
>
> ### [TABLE 1a] Wall-clock Time Comparison of Depth vs. Width Scaling
> |Width 256|Width 512|Width 1024|Width 2048|Width 4096|
> |---|---|---|---|---|
> |4.52 ± 0.34|5.32 ± 0.01|5.96 ± 0.01|8.39 ± 0.02|15.90 ± 0.64|
>
> |Depth 4|Depth 8|Depth 16|Depth 32|Depth 64|
> |-|-|-|-|-|
> |4.52 ± 0.34|5.31 ± 0.43|6.55 ± 0.48|9.14 ± 0.65|14.06 ± 0.66|
>
> ### [TABLE 1b] Scaling Depth Surpasses Performance of Scaling Width in Less Wall-clock time
> |Environment|Width = 4096|Depth = 64|
> |-|-|-|
> |Humanoid|22.30 ± 0.08|**5.66 ± 0.24**|
> |Ant Big Maze|11.70 ± 0.03|**4.04 ± 0.41**|
> |Arm Push Easy|13.69 ± 1.92|**2.38 ± 0.18**|
>
> ### [TABLE 2a] Successful Depth Scaling on GCBC (Antmaze-medium-stitch, OGBench)
> |Depth|2|4|8|16|32|
> |-|-|-|-|-|-|
> |Performance|0.474 ± 0.11|0.572 ± 0.04|0.574 ± 0.02|0.588 ± 0.00|0.634 ± 0.05|
>
> ### [TABLE 2b] Our Architecture is Critical: Depth Scaling on GCBC Fails with Standard Architecture
> |Depth|2|4|8|16|32|
> |-|-|-|-|-|-|
> |Performance|0.532 ± 0.09|0.746 ± 0.01|0.692 ± 0.02|0.698 ± 0.02|0.210 ± 0.21|
>
> ### [TABLE 3a] Wall-clock Time Comparison of Our Method vs. Baselines
> |Method|Humanoid|Ant Big Maze|Ant U4-Maze|Ant U5-Maze|Ant Hardest Maze|Arm Push Easy|Arm Push Hard|Arm Binpick Hard|Humanoid U-Maze|Humanoid Big Maze|
> |-|-|-|-|-|-|-|-|-|-|-|
> |Scaled CRL (Depth 64)|10.99|11.66|11.43|46.40|23.96|19.52|18.53|29.64|46.74|50.33|
> |SAC|0.46|1.55|1.16|5.73|4.33|8.32|8.49|20.70|3.04|8.55|
> |SAC+HER|0.55|1.63|1.26|6.10|4.51|8.54|8.60|20.74|3.48|9.27|
> |TD3|0.79|1.68|1.30|6.18|5.04|8.41|8.31|18.40|5.35|7.50|
> |GCSL|0.42|1.48|0.70|2.81|2.13|6.37|5.22|7.95|3.07|5.08|
> |GCBC|0.61|1.40|1.11|5.56|4.38|8.28|7.36|16.24|7.18|11.37|
>
> ### [TABLE 3b] CRL Surpasses SAC Performance in Less Wall-clock Time in 7 of 10 Environments
> |Algorithm|Humanoid|Ant Big Maze|Ant U4-Maze|Ant U5-Maze|Ant Hardest Maze|Arm Push Easy|Arm Push Hard|Arm Binpick Hard|Humanoid U-Maze|Humanoid Big Maze|
> |---|---:|---:|---:|---:|---:|---:|---:|---:|---:|---:|
> |**SAC**|**27.8**|92.7|69.8|343.8|259.8|499.2|509.7|1242.1|**182.6**|**512.9**|
> |**Scaled CRL (Depth 64)**|382.2|**0.0**|**0.0**|**0.0**|**27.2**|**114.7**|**398.8**|**266.0**|N/A*|N/A*|
>
> [1] Park, S., et al., ICLR 2025, OGBench: Benchmarking Offline Goal-Conditioned RL, https://arxiv.org/abs/2410.20092
> [2] Lee, H., et al., ICLR 2025,  SimBa: Simplicity Bias for Scaling Up Parameters in Deep Reinforcement Learning. http://arxiv.org/abs/2410.09754
> [3] Lee, H., et al., ICML 2025. Hyperspherical Normalization for Scalable Deep Reinforcement Learning. http://arxiv.org/abs/2502.15280
> [4] Farebrother, J., et al., ICML 2024, Stop Regressing: Training Value Functions via Classification for Scalable Deep RL. http://arxiv.org/abs/2403.03950
> [5] Nauman, M., et al.,  NeurIPS 2024. Bigger, Regularized, Optimistic: Scaling for compute and sample-efficient continuous control. http://arxiv.org/abs/2405.16158
> [6] Laskin, M., et al. ICML 2020, CURL: Contrastive Unsupervised Representations for Reinforcement Learning. https://proceedings.mlr.press/v119/laskin20a.html

---

> > ### Author Response · Authors · 2025-08-05
> >
> > Dear Reviewer,
> >
> > We have worked hard to incorporate the review feedback by running new experiments and revising the paper. Do the revisions and discussions above address your questions? We would greatly appreciate your engagement.
> >
> > Thanks!
> >
> > The Authors

---

> ### Comment · Reviewer_gFHa · 2025-08-05
>
> I appreciate the clarifications and the detailed computation analysis and scaling based analysis. The authors' response has addressed all my concerns, and I will maintain my already high judgement score for the paper.

---

### Decision · Program_Chairs · 2025-09-17

**Decision:**

Accept (oral)

**Comment:**

This paper received positive and enthusiastic reviews from all reviewers. The AC agrees with and shares this enthusiasm.

This paper presents a reinforcement learning paradigm to effectively train (very) deep neural networks with self-supervised reinforcement learning. Using this framework, it then presents an analysis demonstrating that, like in other areas of self-supervised learning, self-supervised RL too can scale effectively with network depth and more and more sophisticated capabilities emerge as depth increases.

The paradigm presented in this paper is relatively straightforward and makes use of three things. First, it uses a simple self-supervised RL algorithm, contrastive RL (CRL), second it makes use of GPU-accelerated RL simulators to collect a large amount of data, and third it makes use of modern network designs. This simplicity and the included code will make it easier for the community to build upon this work.

The results and analysis are both equally impressive. On the JaxGCRL benchmarks the performance of contrastive RL improves by 2x to 50x and their scaled CRL outperforms all other methods (by up to an order of magnitude) in nearly all environments. Their analysis is extensive and well done. While there is more than can be summarized here, the AC would like to draw attention to:
1. their demonstration of new behaviors being discovered at greater depth,
2. the analysis showing the importance of batch size scaling for deeper networks in CRL,
3. and the analysis showing that deeper networks enable the policy to stitch together its training experience of short tasks into handling long tasks as evaluation time.

Overall, this paper presents a paradigm that others can easily build on, impressive results, and analysis that will improve the communities understanding of scaling self-supervised RL. Given this, an Oral presentation is appropriate.